# THINK SOCIALLY VIA COGNITIVE REASONING

## ABSTRACT

LLMs trained for logical reasoning excel at step-by-step deduction to reach verifiable answers. However, this paradigm is ill-suited for navigating social situations, which induce an interpretive process of analyzing ambiguous cues that rarely yield a definitive outcome. To bridge this gap, we introduce Cognitive Reasoning, a paradigm modeled on human social cognition. It formulates the interpretive process into a structured cognitive flow of interconnected cognitive units (e.g., *observation* or *attribution*), which combine adaptively to enable effective social thinking and responses. We then propose CogFlow, a complete framework that instills this capability in LLMs. CogFlow first curates a dataset of cognitive flows by simulating the associative and progressive nature of human thought via tree-structured planning. After instilling the basic cognitive reasoning capability via supervised fine-tuning, CogFlow adopts reinforcement learning to enable the model to improve itself via trial and error, guided by a multi-objective reward that optimizes both cognitive flow and response quality. Extensive experiments show that CogFlow effectively enhances the social cognitive capabilities of LLMs, and even humans, leading to more effective social decision-making. Our repository will be released at: `https://anonymous.4open.science/r/CogFlow2025`.

## 1 INTRODUCTION

Social cognition, the core mental process of human social intelligence, governs how individuals perceive, interpret, and respond to social situations (Fiske & Taylor, 2020). This unique ability allows humans to navigate complex social dynamics wisely (Thorndike, 1920). As large language models (LLMs, OpenAI (2024); Guo et al. (2025)) have been taking on more collaborative roles with humans, their capability of social intelligence is being actively examined (Chen et al., 2024; 2025a). Recent studies have revealed promising signs, including evidence of human-like social behaviors (Park et al., 2023) and lobe structure for social skills (Zhou et al., 2025a). Deeper cognitive analysis further suggests that LLMs spontaneously exhibit human-like cognitive features, e.g., reasoning patterns that mimic empathy (Dong et al., 2025), indicating a potential capacity for social cognition.

Despite the potential evidenced in the aforementioned observational studies, improving the social cognitive abilities of LLMs remains underexplored. The root cause lies in the fundamental mismatch between the LLMs' currently implanted reasoning structures and the nature of social intelligence (Moore et al., 2025). Specifically, LLMs excel at complex tasks like math and coding (Shao et al., 2024; Ni et al., 2024), which rely on **step-by-step logical deduction** to arrive at a single verifiable solution (Zheng et al., 2025). In contrast, reasoning in social situations is an **interpretive process** that involves analyzing ambiguous cues that rarely yield a definitive answer (Gandhi et al., 2023; Xu et al., 2025). Not to mention LLMs, even when humans try to apply rigid logic rules to the fluid social domains, they risk falling into "*cognitive rumination*"(Marjanović et al., 2025), which is a state of over-analyzing simple cues, engaging in redundant reasoning cycles, and producing protracted internal monologues that lead to erroneous judgments or delayed responses (as shown in Figure 1). This exposes a pivotal challenge in applying LLMs to social situations, and defining and implementing an effective LLM reasoning paradigm to close this gap is thus urgently needed.

To this end, we pioneer a complete learning framework to instill social cognition into LLM reasoning. Drawing from social cognitive theory (Bandura et al., 1986), we dissect a social cognition process into six core cognitive units that form social thinking: ***Observation, Attribution, Motivation, Regulation, Efficacy, and Behavior***. For example, in the social scene "*choosing the last member for a new robotics team*" shown in Figure 1, one can predict Carlos's action by first observing the

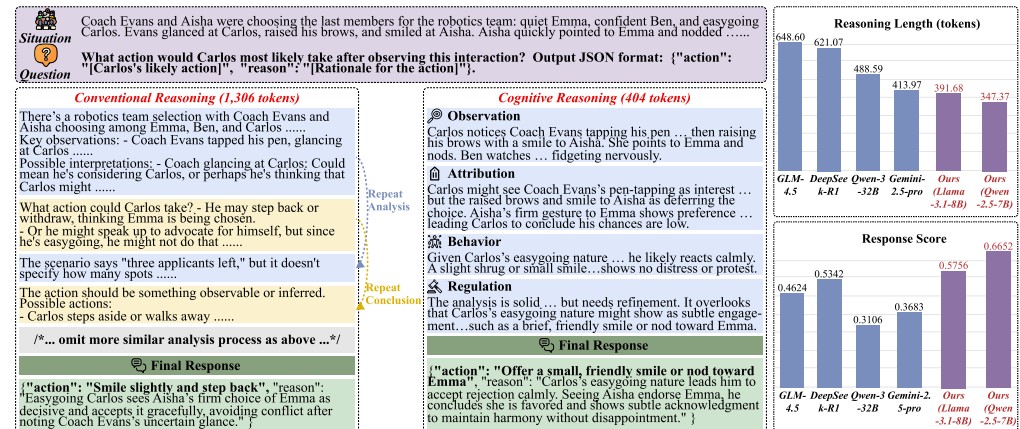

Figure 1: An example of conventional reasoning (DeepSeek-R1) falling into "cognitive rumination", while cognitive reasoning efficiently reaches a better response. The bar chart shows the average reasoning length and comparative preference scores for advanced LLMs on our test set (§4.3).

situation (e.g., *coach Evans smiles at Aisha*), making an attribution about that behavior (e.g., *the smile signals deference*), formulating Carlos's intended behavior (e.g., *a slight shrug*), applying regulation (e.g., *Carlos should show a friendly nod in accordance with his easygoing personality*), and finally leading to predicted actions (e.g., *offer a small, friendly smile or nod toward Emma*). These cognitive units flow adaptively among each other to create an effective, structured reasoning process. We define this process as **Cognitive Reasoning**, a paradigm for thinking and responding effectively in social situations. While cognitive reasoning provides a clear blueprint for social cognition, its implementation in LLMs presents two crucial challenges: **1) Reasoning paradigm shift**: shifting models' training objective from optimizing verifiable logic to guiding analytical reasoning that lacks definitive answers; **2) Cognitive flow control**: teaching the model to adaptively regulate its use of these cognitive units to avoid rumination.

To address these challenges, we teach LLMs to think socially in a form of cognitive flow. **First**, we curate a cognitive reasoning dataset via cognitive flow simulation. We prompt advanced LLMs to simulate human thoughts by crafting cognitive flows about a social situation. This process generates cognitive units sequentially, where each unit acts as a reasoning node that enables the planning of the next, mirroring the associative and progressive process of human cognition. The uncertainty in social situations allows these nodes to naturally branch into a cognitive reasoning tree, and each leaf node contains the response derived from the corresponding cognitive flow about the social situation (as shown in Figure 2). **Second**, given the absence of definitive answers, we design a comparative preference ranking principle to identify the most promising cognitive flows by the relative plausibility of the responses from all leaf nodes. We then prune the flows based on criteria derived from social cognitive theory – coherence, interpretability, predictability (Bandura et al., 1986) – to create high-quality data for supervised fine-tuning (SFT). **Finally**, after instilling basic cognitive reasoning capability via SFT, we empower the model to autonomously explore better reasoning paths using reinforcement learning (RL), guided by a multi-objective reward function: a) a comparative preference reward to steer the model toward flows that yield more plausible responses; and b) a cognitive flexibility reward to encourage adaptive regulation of the cognitive flow's diversity and depth. We name this training framework CogFlow.

Our contributions are summarized as follows: (1) We introduce cognitive reasoning, a pioneering paradigm designed to enable LLMs to think socially via structured interplay among cognitive units. (2) We propose CogFlow, a training framework that instills cognitive reasoning capability into LLM, using a combination of preference-based SFT and multi-objective RL. (3) We conduct extensive experiments showing that CogFlow effectively enhances the social cognitive capabilities of both LLMs and humans, leading to more effective social decision-making.

## 2 Preliminaries

**Definition of Cognitive Reasoning** Humans' social cognition is a dynamic process (Fiske & Taylor, 2020) where people navigate complex social situations by building and refining internal cognitive

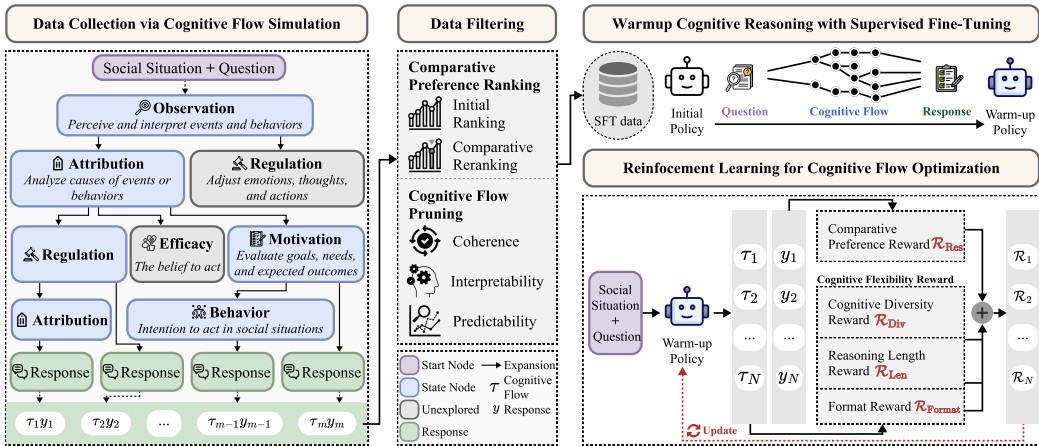

Figure 2: Overview of our CogFlow framework, which crafts cognitive flows via tree-structured planning, uses the filtered data for SFT, and then employs multi-objective RL for self-improvement.

maps (Tolman, 1948). This occurs in a feedback loop: people map social situations to their actions (Bandura & Walters, 1977), and the outcomes provide feedback that reshapes both their internal map and the external situations (Bandura et al., 1986). We formalize the mental activity within this loop as *cognitive reasoning*, and operationalize the cognitive map as a "cognitive flow", an adaptive adoption of several core cognitive units (Bandura et al., 1986), e.g., attributions about a person's intent shape one's motivation to interact, and a strong sense of efficacy can enhance regulation strategies. Definitions of the cognitive units are: (1) `Observation`: Perceiving and interpreting events and others' behaviors to form an initial cognitive appraisal of a situation (Lazarus, 1991). (2) `Attribution`: Analyzing the causes of events or behaviors (Heider, 2013). (3) `Motivation`: The expectations and value assessments of potential outcomes for self and others' behavior, which provide the drive to act (Vroom, 1964). (4) `Regulation`: Reflecting on and adjusting emotions, beliefs, and behaviors in pursuit of social goals (Carver & Scheier, 2012). (5) `Efficacy`: The belief to execute a specific social behavior, which influences motivational intensity (Bandura, 1997). (6) `Behavior`: Formulating an intention to act in response to social situations (Ajzen, 1991).

**Task Formulation in Cognitive Reasoning** Given a social situation $\mathcal{S}$ and a query $\mathcal{Q}$, the goal is to obtain a response $y$ by first generating an explicit cognitive flow $\tau$. Each reasoning step is materialized by a particular cognitive unit $r_i = (u_i, c_i)$, where $u_i$ is the unit category (e.g., `Observation`) and $c_i$ is the materialized text content of $u_i$. A complete flow $\tau$ is thus an ordered sequence of $n$ reasoning steps: $\tau = \{r_1, r_2, \cdots, r_n\} = \{(u_1, c_1), (u_2, c_2), \cdots, (u_n, c_n)\}$. We define the input $x$ as the concatenation of $\mathcal{S}$ and $\mathcal{Q}$, $x = [\mathcal{S}; \mathcal{Q}]$. Our goal is to learn a policy $\pi_\theta$ that maximizes the joint probability of generating the flow $\tau$ and the response $y$: $\pi_\theta(\tau, y|x) = \pi_\theta(y|\tau, x) \cdot \pi_\theta(\tau|x)$, where $\pi_\theta(\tau|x)$ depicts the generation of structured cognitive flow $\tau$, $\pi_\theta(y|\tau, x)$ measures the correspondence between cognitive reasoning content of $\tau$ and produced response $y$ about input $x$.

## 3 METHODOLOGY

As shown in Figure 2, our training framework CogFlow begins by collecting cognitive flows via tree-structured planning, guided by carefully crafted instructions. We implement a dual-validated data filtering procedure: a comparative preference ranking module identifies cognitive flows that yield high-quality responses, which are then pruned under coherence, interpretability, and predictability criteria. After instilling such structured cognitive reasoning into an LLM via SFT, the model improves itself via RL, guided by a multi-objective reward that optimizes both cognitive flow and response quality. The instructions used in data collection and more details are shown in App. C.

### 3.1 DATA COLLECTION VIA COGNITIVE FLOW SIMULATION

**Seed Data Collection** To approach realistic and complex social situations, we collect seed data from Reddit. Unlike existing datasets, e.g., SocialIQA (Sap et al., 2019), which often feature simple situations with limited social dynamics, our collection focuses on complex multi-person interactions, presenting a more substantial challenge for LLMs. The pipeline is constructed as follows:

- **Situation curation**: We curate and anonymize Reddit posts, removing all sensitive content. Then we prompt Deepseek-R1 (hereafter R1) to distill them into concise situation descriptions ($\mathcal{S}$).
- **Question generation**: For each situation, we prompt R1 to extract detailed social cues and generate a corresponding question ($\mathcal{Q}$) that demands deep interpretation, analysis, and prediction.
- **Filtering**: To ensure high quality of our seed data, each generated situation-question pair ($\mathcal{S}$,$\mathcal{Q}$) is re-assessed by R1. We discard pairs rated low on situation complexity or question relevance.

**Cognitive Flow Simulation** To obtain human-like cognitive flows, we simulate the associative and progressive process of human cognition (Bandura et al., 1986). The flows are crafted by prompting R1 to sequentially plan and materialize the content of cognitive units. Each unit acts as a reasoning node, and the preceding path supports planning for the next. As a single thought can lead to multiple continuations, this process naturally forms a tree-structured exploration of reasoning flows.

- **State**: Each node in the tree is a state $s_k$, denoting a partially generated cognitive reasoning path from the root. The root node, $s_0$, corresponds to the initial input $x = [S; Q]$.
- **Action**: At any state $s_k$, we prompt R1 to choose the next cognitive unit $u_{k+1}$ from dynamic candidates, $A(s_k) \subseteq \{u_1, \cdots, u_6\}$. The initial state $s_0$ is constrained to be unit `Observation`.
- **Planning**: Planning begins from the root state $s_0$ and iteratively expands the tree by: *a) Generation*: For the selected unit $u_{k+1}$, we prompt R1 to generate the unit's text content $c_{k+1}$ with respect to the current state $s_k$. This forms a new reasoning node $r_{k+1} = (u_{k+1}, c_{k+1})$ and expands the path to a new state $s_{k+1} = [s_k; r_{k+1}]$. *b) Prediction*: R1 is then prompted to analyze the reasoning path $s_{k+1}$ and predict a set of relevant next cognitive units $A(s_k)$ to explore. This step adaptively prunes the action space from all six possible units to only the most contextually appropriate ones. *c) Expansion*: Each candidate unit from the predicted set becomes a new node, expanding the tree with multiple parallel reasoning paths.
- **Completion**: This expansion process repeats until R1 determines that the reasoning has reached a terminal state, which is referred to as a leaf node. At this point, it generates a final response $y$ based on the fully constructed chain.

We define a complete cognitive flow from the root node to any leaf node as a rollout. By performing multiple rollouts for each seed instance, we collect a diverse set of cognitive flows $\{\tau_1, \cdots, \tau_m\}$ and their corresponding final responses $\{y_1, \cdots, y_m\}$.

**Dual-Validation based Filtering** In the absence of definitive answers, we design a two-step filtering procedure to ensure the quality of generated cognitive flows. We first identify flows landing on high quality responses via **two-stage Comparative Preference Ranking** (*CPRank*[2]):

- **Comparison pool construction**: For each seed instance, we craft a candidate pool containing responses from our rollouts and a baseline response directly from R1. The pool size is set to 10, which we found to be satisfactory in our preliminary tests. If the size of valid rollouts is fewer than 10, we create variations by perturbing the generated flows (e.g., combining flow snippets).
- **Initial ranking**: We prompt R1 to generate situation-specific criteria and then use them to assign an initial score and critique for each response in the pool, i.e., LLM-as-a-judge (Liu et al., 2025).
- **Comparative reranking**: To mitigate scoring biases (e.g., positional bias in R1's initial scores), we then select the median-ranked response as an anchor. We ask R1 to perform a final comparative reranking of the entire pool against this anchor, yielding a more robust preference order.

Next, we conduct **cognitive flow pruning**. We select flows with responses scored higher than those from R1, designating them as high-quality candidates. The candidates are then pruned by R1 using the following criteria constructed based on social cognitive theories (Bandura et al., 1986): *a) coherence*: logically sound and free of contradictions; *b) interpretability*: clearly explain the social dynamics; *c) predictability*: offer reasonable insight into the future evolution of social dynamics. Only cognitive flows satisfying all criteria are retained.

### 3.2 WARMUP COGNITIVE REASONING WITH SFT

To endow LLM with basic cognitive reasoning capability, we train it with the constructed cognitive flows via SFT. For each curated data instance $(x, \tau, y)$, we format it by concatenating all reasoning steps $r_i = (u_i, c_i)$ within the cognitive flow $\tau$ into a continuous text sequence $\tau_{\text{SFT}}$: $\tau_{\text{SFT}} = \oplus_{r_i \in \tau} \langle u_i \rangle c_i \langle /u_i \rangle$, where $\oplus$ is string concatenation. The cognitive unit tags $\langle u_i \rangle$ and $\langle /u_i \rangle$ (e.g., $\langle \text{Observation} \rangle$ and $\langle /\text{Observation} \rangle$) are added to LLM's vocabulary as new special tokens, allowing them to be directly embedded and enabling the LLM to learn cognitive reasoning structure intrinsically. Policy $\pi_\theta$ is optimized by minimizing the standard SFT loss:

$$\mathcal{L}_{\text{SFT}}(\theta) = -\mathbb{E}_{(x,\tau,y) \sim \mathcal{D}_{\text{SFT}}} [\log(\pi_\theta(\tau_{\text{SFT}}, y | x)], \tag{1}$$

where $\mathcal{D}_{\mathrm{SFT}}$ is our curated data for SFT. After warmup, $\pi_\theta$ is able to generate structured cognitive flows using these specialized tags without relying on manually crafted prompts.

### 3.3 REINFORCEMENT LEARNING FOR COGNITIVE FLOW OPTIMIZATION

To enable the model to progressively refine its cognitive flows, we adopt RL, where the policy model learns to improve its cognitive reasoning via trial and error. We use GRPO (detailed in Appendix B, Shao et al. (2024)) to optimize policy $\pi_\theta$, which is guided by our multi-objective reward as follows:

**Comparative Preference Reward** ($\mathcal{R}_{\mathrm{Res}}$)    While the preference ranking used for data filtering is well-suited for scenarios lacking definitive answers, it is not economically feasible for large-scale online training (which needs to execute R1 against each generated rollouts). We thus train a dedicated reward model $RM_\phi$ to predict pairwise preference. For each input $x$, $RM_\phi$ is trained to predict whether a candidate response $y$ is preferred over a set of $k$ reference responses $\{y_{\mathrm{ref}}^1, \cdots, y_{\mathrm{ref}}^k\}$ (we use $k = 3$). During RL, for each generated response $y$, we use the top-$k$ responses from our curated data as the reference set and set the reward to be $RM_\phi$'s predicted probability of $y$ is preferred:

$$\mathcal{R}_{\mathrm{Res}}(y|x) = P_\phi(y \succ \{y_{\mathrm{ref}}^1, \cdots, y_{\mathrm{ref}}^k\}|x). \tag{2}$$

**Cognitive Flexibility Reward**  Beyond response, we foster policy $\pi_\theta$ to regulate its thought process.

- **Cognitive diversity reward** ($\mathcal{R}_{\mathrm{Div}}$): To prevent the model from falling into simplistic or repetitive reasoning patterns, we introduce $\mathcal{R}_{\mathrm{Div}}$ to encourage exploration of diverse cognitive flows. This design is inspired by human social cognition, where people flexibly adapt their cognitive strategies to situational nuances (Fiske & Taylor, 2020). The reward evaluates a cognitive flow $\tau$ by incentivizing the use of rarer cognitive units within a batch of rollouts, encouraging the model to avoid over-reliance on common reasoning steps. Given $m$ rollouts for an input $x$, yielding flows $\{\tau_1, \ldots, \tau_m\}$, the reward for a chain $\tau$ containing $v$ unique cognitive units $\{u_1, \ldots, u_v\}$ is:

$$\mathcal{R}_{\mathrm{Div}}(\tau) = -\frac{1}{v} \sum_{j=1}^{v} \log(p(u_j)), \tag{3}$$

  where $p(u_j)$ is the frequency of the cognitive unit $u_j$ across all $m$ sampled cognitive flows.

- **Reasoning length reward** ($\mathcal{R}_{\mathrm{Len}}$): While cognitive diversity is crucial, it must be balanced with conciseness to avoid cognitive rumination, i.e., overly long and unproductive reasoning. We therefore introduce $\mathcal{R}_{\mathrm{Len}}$ to encourage focused yet comprehensive thought by penalizing cognitive flows that are either too short or too long. For each input $x$, we build a dynamic target length range $[L_{\min}, L_{\max}]$ derived from the top-$k$ reference flows in our curated data. The reward is calculated using a soft bounding function created by multiplying two sigmoid functions. This forms a "reward window" that gently penalizes flows whose length is outside the desired length range:

$$\mathcal{R}_{\mathrm{Len}}(\tau) = \sigma\left(\frac{|\tau| - (L_{\min})/2}{L_{\min}/8}\right) \cdot \sigma\left(\frac{L_{\max} + L_{\min} - |\tau|}{L_{\max}/8}\right) \tag{4}$$

- **Structural format reward** ($\mathcal{R}_{\mathrm{Format}}$): To maintain structural integrity, we use a rule-based binary reward to encourage the cognitive flows to follow the required $\langle u_i \rangle c_i \langle /u_i \rangle$ structure:

$$\mathcal{R}_{\mathrm{Format}}(\tau) = \begin{cases} 1 & \text{if format of } \tau \text{ is valid} \\ 0 & \text{otherwise} \end{cases} \tag{5}$$

**Weighted Reward Function**    The final reward for a rollout $(\tau, y)$ is a weighted combination of the above rewards, with the format reward acting as a gate to discard structurally invalid flows:

$$\mathcal{R} = \mathcal{R}_{\mathrm{Format}} \cdot (\omega_1 \cdot \mathcal{R}_{\mathrm{Res}} + \omega_2 \cdot \mathcal{R}_{\mathrm{Div}} + \omega_3 \cdot \mathcal{R}_{\mathrm{Len}}). \tag{6}$$

## 4 EXPERIMENTS

### 4.1 EXPERIMENTAL SETUP

**Datasets**    We collected 5,100 social situations from Reddit, spanning 5 major categories and 16 subcategories; and each post passed rigorous safety filtering (Kim et al., 2023). Each seed instance yielded an average of 3.6 high-quality cognition flows, with each flow containing 4 cognitive units on average. To validate data quality, we employ six domain experts (with Master's degrees or higher) to inspect 500 random instances, resulting in a 96.8% pass rate that confirms the dataset's satisfying quality. For model training, we allocate 1,000 seed instances with 3,661 cognitive flows for SFT and 3,600 instances for RL (3,200 for training and 400 for validation). Another 500 instances are used for the final evaluation. Moreover, we extract 26,676 candidate-reference response pairs to build our comparative preference reward model. More details of our dataset are provided in Appendix D.1.

Table 1: Results of consistency between evaluators and human judgments. The agreement ratio *kappa* $\in [0.41, 0.6]$ denotes moderate agreement.

| Evaluators | Easy | Medium | Hard | Overall |
|---|---|---|---|---|
| Score-R1 | 0.5604 | 0.4938 | 0.4848 | 0.5141 |
| Score-Q32B | 0.5824 | 0.5219 | 0.5152 | 0.5405 |
| CPRank-Q32B | 0.6374 | 0.5094 | 0.6515 | 0.5669 |
| CPRank-R1 | 0.6374 | 0.5625 | 0.4697 | 0.5757 |
| $RM_\phi$ | 0.5714 | 0.5781 | 0.6667 | 0.5863 |
| $CPRank^2$-Q32B | 0.6648 | 0.6031 | 0.5909 | 0.6215 |
| $CPRank^2$-R1 | 0.6538 | 0.6312 | 0.6212 | 0.6373 |
| *kappa* | 0.4534 | 0.4559 | 0.5041 | 0.4693 |

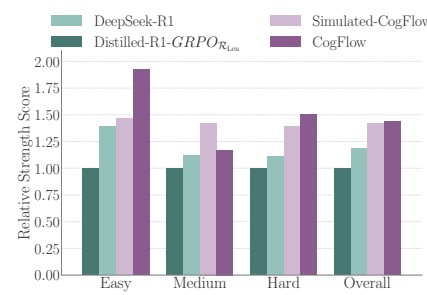

Figure 3: Pairwise results from Bradley-Terry model. The higher strength score, the better.

**Baselines and LLM Evaluators**    We compared baselines: **(1) Tuning-free Reasoning LLMs**: OpenAI o3, o3-mini, GLM-4.5 (GLM et al., 2025), Qwen-3-32B (Yang et al., 2025), DeepSeek-R1, Gemini-2.5-pro/flash. **(2) Fine-tuned Open-source LLMs**: we trained Llama-3.1-8B (Meta, 2024) and Qwen-2.5-7B (Yang et al., 2024) backbones for CogFlow and baselines: a) *Direct-SFT/GRPO*: backbones trained directly on the responses in our curated dataset without cognitive flow, where GRPO relies solely on our $\mathcal{R}_{Res}$ reward (same use below). b) *Distilled-R1-SFT/GRPO/GRPO$_{\mathcal{R}_{Len}}$*: backbones trained on distilled R1's reasoning chains by SFT, GRPO, and GRPO using both $\mathcal{R}_{Res}$ and $\mathcal{R}_{Len}$. c) *CogFlow-SFT/GRPO*: backbones trained on our data with cognitive flows. We use R1 and cost-effective Qwen-3-32B as our LLM evaluators to perform two-stage comparative preference ranking, called *CPRank$^2$-R1/Q32B*. For each test instance, a model's generated response is ranked within a comparison pool containing the pre-curated reference responses. The rank is normalized to a score by $(M - rank)/(M - 1)$, M=10. A model's performance is its average score across all test instances. More details about our baselines and evaluators are reported in Appendix D.2 and D.3.

## 4.2 Main Results by Human Evaluations

**LLM Evaluators' Consistency with Human Judgment**    We evaluate 7 LLM evaluators: 1) *CPRank$^2$-Q32B&R1* and their variation without reranking (denoted as *CPRank-Q32B&R1*), 2) reward model $RM_\phi$ trained on Qwen-2.5-7B, and 3) prompt-based direct scoring baselines (denoted as *Score-Q32B&R1*). Six experts perform pairwise comparisons on all 500 seed instances in our test set, each with 4 distinct responses from 4 models: *CogFlow* (trained on Llama, same use below experiments), *Distilled-R1-GRPO$_{\mathcal{R}_{Len}}$*, *DeepSeek-R1*, and *Simulated-CogFlow* (top-ranked response in cognitive flow simulation). To balance workload, 500 instances are split into two sets, each assigned to 3 experts. Experts provide *win/tie/loss* judgments for all response pairs and label difficulty of each instance (*easy, medium, hard*). The final preference label of each pair and the difficulty label of each seed instance are determined by majority vote among human experts. We measure the consistency between the pairwise orderings of LLM evaluators and the aggregated human judgments.

The results in Table 1 reveal: (1) two-stage ranking (*CPRRank$^2$-R1&Q32B*) aligns best with human judgment, achieving the highest overall consistency. Against their single-stage counterparts, the reranking step is crucial for improving alignment by mitigating initial scoring biases. (2) our trained reward model is an effective proxy. $RM_\phi$ outperforms all direct scoring and single-stage ranking baselines, showing it is a cost-effective substitute for expensive LLM judges during online training.

**Results of Pairwise Comparison**    We convert experts' pairwise *win/tie/loss* judgments into scalar scores using the Bradley-Terry model (Bradley & Terry (1952), detailed in Appendix D.6). The results in Figure 3 show that *CogFlow* surpasses its teacher model (*DeepSeek-R1*), showing our framework enables a smaller model to internalize cognitive reasoning to produce high-quality responses effectively. More importantly, *CogFlow* performs on par with *Simulated-CogFlow*, while *Distilled-R1-GRPO$_{\mathcal{R}_{Len}}$* remains inferior to its teacher. This clearly suggests that cognitive reasoning is consistently beneficial for social responding, and cognitive reasoning is effectively learnable.

**Helpfulness of Cognitive Flow for Humans**    Beyond evaluating LLMs, we assess the utility of cognitive flow for humans, including its quality and value as a cognitive aid for human decision-making. We hired another 6 annotators to assess 100 multiple-choice instances from the experts' curated dataset. For each instance, the golden response from expert consensus is explained by one of two reasoning styles: our *cognitive flow* or *R1 reasoning chain*, shown in a balanced frequency. Annotators are evenly split to perform two tasks: *(1) Quality ratings* with a 1-5 scale on *coherence*,

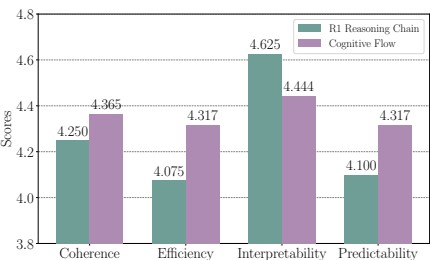 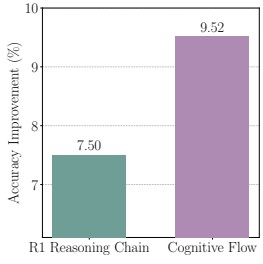 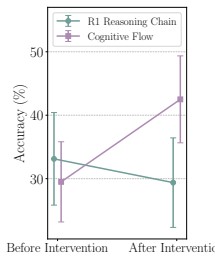

(a) Results of reasoning quality rated by humans with four criteria on a 1-5 scale.

(b) Results of helpfulness for human decision-making.

(c) Results of cognitive intervention for humans.

Figure 4: Quantitative analysis of two types of reasoning (cognitive reasoning vs. R1 reasoning).

*efficiency* (conciseness of content and logic), *interpretability*, and *predictability*; *(2) Value test*: Annotators first select their preferred response from four options and are then shown the reasoning process behind the experts' preferred response. Finally, they are asked to make their final decision. "*Helpfulness*" is the average accuracy improvement before and after exposing the reasoning content.

The results in Figure 4a and 4b reveal that cognitive flows are better received by the annotators, as they are considered more coherent, efficient, and have higher predictability, making them better tools to understand social situations. We can also notice that there is a trade-off between interpretability and efficiency: R1's reasoning often includes exhaustive self-reflection that, while boosting its interpretability by listing all social cues, does so at the cost of lower efficiency. Most importantly, cognitive flows are more helpful for human decision-making, yielding a higher accuracy improvement, showing the potential to augment human social intelligence.

**Cognitive Intervention for Humans**  To further study cognitive reasoning's potential on humans, we conduct a preliminary cognitive intervention trial. We prepare two types of interventions: **1)** *Cognitive flow-style guidance*: emphasize key social cues and analytical steps among cognitive units to guide humans' social thinking. **2)** *R1 reasoning chain-style guidance*: provide a chain-of-thought summary. To create the guidance, we prompt R1 to convert the cognitive flow/R1 reasoning chain into hints that illuminate the thought process without revealing the final answer (see Appendix D.7 for examples). We recruit 20 volunteers who are randomly assigned to an experimental group (*cognitive flow-style*) or a control group (*R1 reasoning chain-style*). Before intervention, each participant completes 10 tasks without guidance, and then an ANOVA test (Fisher, 1970) confirms that no statistically significant differences exist between the two groups ($p = 0.42$). During the intervention, participants sequentially complete 25 instances without guidance and 25 with guidance, allowing us to measure the change in decision-making accuracy due to the intervention. Results in Figure 4c show the ***cognitive flow*-style intervention significantly improves participants' social decision-making accuracy**, while the *R1 reasoning chain*-style shows a slight downward trend. This reveals the potential of structured cognitive reasoning to improve humans' ability for social thinking.

### 4.3 RESULTS ON AUTOMATED EVALUATIONS

**Main Results**  For each model, we generate 4 responses per test instance and average the scores in Table 2. Results reveal *CogFlow* outperforms all baselines on both backbone models and evaluators, showing the effectiveness of combining structured cognitive reasoning with RL. Cognitive reasoning has a clear edge to unstructured reasoning in improving model learning, e.g., *CogFlow* vs. *Distilled-R1-GRPO$_{\mathcal{R}_{\text{Len}}}$*. Besides, across both model family and reasoning style, models tuned with RL clearly outperform their SFT counterparts, showing RL's ability to effectively refine the reasoning strategies. Another finding is that models trained on cognitive flows produce significantly shorter yet more effective reasoning, showing the capability of cognitive reasoning to reduce reasoning costs.

**Ablation Study of Rewards**  Results in Table 2 reveal: (1) $\mathcal{R}_{\text{Div}}$ promotes exploration but requires constraints. When $\mathcal{R}_{\text{Len}}$ is removed, performance drops sharply and reasoning length nearly doubles (e.g., 391.68 vs. 725.77 tokens on Llama). This shows while $\mathcal{R}_{\text{Div}}$ successfully encourages diverse cognitive flow, it leads to inefficient reasoning if left unconstrained. (2) $\mathcal{R}_{\text{Len}}$ ensures reasoning efficiency and quality. When $\mathcal{R}_{\text{Div}}$ is removed, performance remains higher than GRPO baseline, while reasoning length is effectively controlled (e.g., 314.03 vs. 282.73 tokens on Llama). This shows $\mathcal{R}_{\text{Len}}$ acts as a vital regularizer, guiding the model toward concise and high-quality reasoning.

Table 2: Automatic evaluation results. The best results in the two model families are **bold**.

| Models | CPRank$^2$-R1 ($\uparrow$) | | | | CPRank$^2$-Q32B ($\uparrow$) | | | | Reasoning Length (tokens, $\downarrow$) |
|---|---|---|---|---|---|---|---|---|---|
| | Overall | Easy | Medium | Hard | Overall | Easy | Medium | Hard | |
| **Tuning-free Reasoning LLMs** | | | | | | | | | |
| o3-mini | 0.2205 | 0.3376 | 0.2053 | 0.1185 | 0.3140 | 0.4117 | 0.3103 | 0.2189 | 507.87 |
| o3 | 0.2933 | 0.4191 | 0.2497 | 0.2214 | 0.4096 | 0.4567 | 0.4106 | 0.3659 | 163.08 |
| Qwen3-32B | 0.3106 | 0.3696 | 0.3709 | 0.1912 | 0.4243 | 0.5184 | 0.3893 | 0.3875 | 488.59 |
| Gemini-2.5-flash | 0.3111 | 0.4316 | 0.2557 | 0.2460 | 0.4383 | 0.4713 | 0.4419 | 0.3977 | 360.42 |
| Gemini-2.5-pro | 0.3683 | 0.4347 | 0.2818 | 0.3883 | 0.5037 | 0.5525 | 0.4862 | 0.4835 | 413.97 |
| GLM-4.5 | 0.4624 | 0.4967 | 0.4203 | 0.4702 | 0.5663 | 0.5208 | 0.5519 | 0.6395 | 648.60 |
| DeepSeek-R1 | 0.5342 | 0.5220 | 0.4990 | 0.5816 | 0.6578 | 0.6267 | 0.6485 | 0.7067 | 621.07 |
| **Tuned Llama-3.1-8B-Instruct Series** | | | | | | | | | |
| Direct-SFT | 0.3545 | 0.3962 | 0.3490 | 0.3193 | 0.5407 | 0.6236 | 0.5144 | 0.5011 | - |
| Direct-GRPO | 0.5041 | 0.5154 | 0.5764 | 0.4208 | 0.7196 | 0.7751 | 0.7332 | 0.6380 | - |
| Distilled-R1-SFT | 0.3213 | 0.4530 | 0.2202 | 0.2941 | 0.4508 | 0.5036 | 0.4383 | 0.4181 | 554.76 |
| Distilled-R1-GRPO | 0.5157 | 0.5603 | 0.5601 | 0.4279 | 0.7310 | 0.8127 | 0.7400 | 0.6305 | 568.90 |
| Distilled-R1-GRPO$_{\mathcal{R}_{\text{Len}}}$ | 0.5017 | 0.6167 | 0.4438 | 0.4474 | 0.7519 | 0.8080 | 0.7423 | 0.7108 | 444.90 |
| CogFlow-SFT | 0.4024 | 0.4827 | 0.3443 | 0.3821 | 0.5999 | 0.6472 | 0.5772 | 0.5916 | 451.14 |
| CogFlow-GRPO | 0.5564 | **0.6974** | 0.5501 | 0.3193 | 0.7420 | 0.7534 | 0.7441 | **0.7265** | 314.03 |
| **CogFlow** (ours) | **0.5756** | 0.6645 | 0.5350 | **0.5294** | **0.7828** | **0.8271** | **0.7908** | 0.7232 | 391.68 |
| CogFlow (w/o $\mathcal{R}_{\text{Len}}$) | 0.5525 | 0.6199 | 0.5665 | 0.4727 | 0.7069 | 0.7271 | 0.7359 | 0.6347 | 725.77 |
| CogFlow (w/o $\mathcal{R}_{\text{Div}}$) | 0.5702 | 0.5783 | **0.6250** | 0.5073 | 0.7574 | 0.8176 | 0.7431 | 0.7202 | 282.73 |
| **Tuned Qwen-2.5-7B-Instruct Series** | | | | | | | | | |
| Direct-SFT | 0.3144 | 0.3451 | 0.3239 | 0.2742 | 0.5113 | 0.5862 | 0.5016 | 0.4500 | - |
| Direct-GRPO | 0.6148 | 0.7147 | 0.6407 | 0.4914 | 0.7630 | 0.8221 | 0.7605 | 0.7062 | - |
| Distilled-R1-SFT | 0.1751 | 0.2083 | 0.1101 | 0.1984 | 0.3776 | 0.4086 | 0.3689 | 0.3606 | 711.80 |
| Distilled-R1-GRPO | 0.5261 | 0.6109 | 0.4682 | 0.5013 | 0.7061 | 0.7171 | 0.7395 | 0.6355 | 955.16 |
| Distilled-R1-GRPO$_{\mathcal{R}_{\text{Len}}}$ | 0.5298 | 0.5727 | 0.5970 | 0.4206 | 0.7458 | 0.8038 | 0.7398 | 0.6962 | 437.06 |
| CogFlow-SFT | 0.3672 | 0.4186 | 0.4032 | 0.2810 | 0.5567 | 0.5838 | 0.5564 | 0.5291 | 368.41 |
| CogFlow-GRPO | 0.5988 | **0.6750** | 0.5871 | 0.5361 | 0.7542 | 0.7971 | 0.7526 | 0.7124 | 237.11 |
| **CogFlow** (ours) | **0.6652** | 0.6404 | **0.7531** | **0.6015** | **0.7956** | **0.8248** | **0.7963** | **0.7641** | 347.37 |
| CogFlow (w/o $\mathcal{R}_{\text{Len}}$) | 0.6142 | 0.6462 | 0.6133 | 0.5840 | 0.7568 | 0.7784 | 0.7610 | 0.7269 | 502.30 |
| CogFlow (w/o $\mathcal{R}_{\text{Div}}$) | 0.6084 | 0.6660 | 0.6249 | 0.5356 | 0.7824 | 0.7902 | 0.7930 | 0.7555 | 277.24 |

(a) Patterns in *CogFlow-SFT*      (b) Patterns in *CogFlow*      (c) Patterns in *CogFlow (w/o $\mathcal{R}_{\text{Div}}$)*

Figure 5: The transition patterns of cognitive units in reasoning. The proportion of units at different depths is denoted by node size, the transition probability between units is denoted by edge thickness.

## 4.4 IN-DEPTH ANALYSIS OF THE COGNITIVE FLOW

**Transition Patterns of Cognitive Units** To dissect the structure of cognitive flow, we plot the frequency of each cognitive unit at different reasoning depths (denoted by node size) and the transition probability between the units (denoted by edge thickness). Figure 5 reveals that: (1) *Cogflow* can guide the model to learn more ordered cognitive strategies. Against diffuse reasoning patterns from *CogFlow-SFT*, *CogFlow* exhibits a more structured and hierarchical cognitive flow, which also leads to higher-quality responses, as supported by Table 2. (2) The diversity reward $\mathcal{R}_{\text{Div}}$ is critical for preventing pattern collapse. *CogFlow (w/o $\mathcal{R}_{\text{Div}}$)* shows a stark collapse into a monotonous and rigid reasoning path (e.g., *Observation→Attribution→Motivation*). This highlights the importance of $\mathcal{R}_{\text{Div}}$ for maintaining cognitive flexibility, supporting the results of ablation study.

**Information Flow within Cognitive Flow** To dissect the internal mechanism of cognitive reasoning, we visualize the information flow within our *CogFlow* during inference. We analyze attention patterns between four logical blocks: initial **input**, **cognitive unit tokens** (e.g., $\langle$Observation$\rangle$), **cognitive unit content** (the text generated for each unit), the final **response**. The attention weight between blocks is calculated by averaging the summed weights from all layers. For multi-token blocks (*input, content, response*), we mitigate dilution from non-essential tokens by averaging the top-10 attention weights within a block. The resulting weights ($w$) are then normalized ($w' = w^{0.2}$) to enhance the visibility of all connections. For clarity, we separate this information flow into 3 patterns: *unit-to-unit*, *unit-to-content*, and *content-to-content*. A flow, shown in Figure 6, reveals: **(1) The structural unit-to-unit flow dominates the reasoning process**, with the highest attention weights

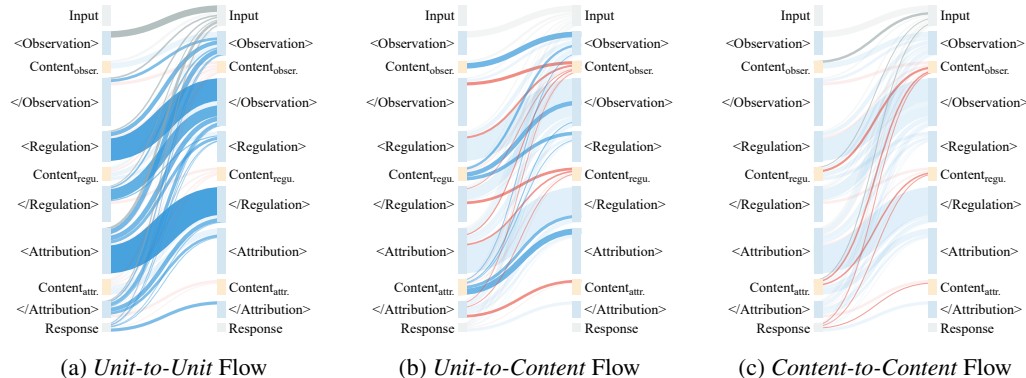

(a) *Unit-to-Unit* Flow      (b) *Unit-to-Content* Flow      (c) *Content-to-Content* Flow

Figure 6: Visualization of three information flow patterns (a-c) in performing cognitive reasoning. The patterns are composed of cognitive units (blue blocks) and their unit content (yellow blocks). The connecting line width reflects the degree of influence from the right block to the left.

among all patterns. **(2) Cognitive unit tokens actively steer content generation**. The unit-to-content flow shows a strong link between a unit token and its unit content (e.g., $Content_{attribution}$ to $\langle Attribution \rangle$), confirming that the unit tokens are not just placeholders but actively guide the generation of relevant thoughts. **(3) Reasoning exhibits a hierarchical "structure-first" attention flow.** When generating new content, the model consistently attends more to the structural tokens of previous steps than to the textual content of those steps, e.g., $Content_{regulation}$ attends more strongly to the $\langle /Observation \rangle$ token than to the $Content_{observation}$. This shows that the structured scaffold built by unit tokens is the primary driver of the model's cognitive flow generation.

## 5 RELATED WORK

Recent works have revealed parallels between LLMs and human social behavior (Park et al., 2023; Zhou et al., 2023; 2025b), social skills (Zhou et al., 2025a), and deeper cognitive habits in reasoning (Dong et al., 2025), inspiring ideas to integrate cognitive theories (Chen et al., 2024; 2025a) like simulation theory (Wilf et al., 2024; Sarangi et al., 2025), to guide LLM reasoning (Wang et al., 2024a; AlKhamissi et al., 2025). Yet, most methods rely on prompting to enforce specific strategies (Wang et al., 2024a; Park et al., 2025) While externally shaping an LLM's reasoning, it brings superficial mimicry of prompt's format instead of instilling adaptive reasoning (Zhou et al., 2023).

A practical solution is to internalize reasoning into LLM's parameters via training (Magister et al., 2023; Paliotta et al., 2025), adapting prompt-based CoT (Wei et al., 2023; Yao et al., 2023) for reasoning models (OpenAI, 2024; Guo et al., 2025). This paradigm has proven effective for tasks like math (Shao et al., 2024), which rely on step-by-step logical deductions to reach verifiable outcomes (Zheng et al., 2025). Yet, it can induce over-thinking (Chen et al., 2025b; Kumar et al., 2025; Cuadron et al., 2025), a state of repetitive thought cycling (Gandhi et al., 2025), prompting efforts to improve efficiency (Wang et al., 2024b; 2025). More critically, such logical reasoning is ill-suited for social situations, which involve an interpretive process of analyzing ambiguous cues that rarely yield a definitive answer (Gandhi et al., 2023; Xu et al., 2025; Moore et al., 2025). Applying this deductive paradigm to the fluid social domain risks "cognitive rumination", i.e., over-analysis simple social cues (Marjanović et al., 2025). Thus, we introduce cognitive reasoning to bridge this gap.

## 6 CONCLUSIONS

In this paper, we introduce cognitive reasoning, a paradigm that models human social cognition by formulating it into a structured cognitive flow of interconnected cognitive units. We then propose CogFlow, a complete framework that instills the cognitive reasoning capability in LLMs using a combination of preference-based SFT and multi-objective RL. Our extensive experiments show that CogFlow significantly enhances the social cognitive capabilities of LLMs, leading to more effective social decision-making. Furthermore, our findings from the human intervention trial reveal that the structured cognitive flow also holds promise as a tool for augmenting human social intelligence.

ETHICS STATEMENT

We have carefully considered the ethical implications of our work throughout the entire research process, from data collection to human evaluation and potential societal impact.

**Data Sourcing and Privacy**    The seed data for our research was sourced from public Reddit posts. To uphold the principle of respecting privacy and avoiding harm, we implemented a strict data processing pipeline. This pipeline included (1) the complete anonymization of all posts, removing any personally identifiable information, usernames, or sensitive content, and (2) the application of rigorous safety filters as described by (Kim et al., 2023) to eliminate potentially harmful or offensive content. The resulting dataset consists of distilled, non-personal social situations intended solely for academic research.

**Human Participant Engagement**    Our study involved human participation in several evaluation stages: 6 domain experts (different individuals with a master's degree or higher) for data validation, 6 annotators for evaluating reasoning chains, and 20 volunteers for a cognitive intervention trial. For all human-involved experiments, we adhered to the following: (1) All participants were fully informed about the nature and purpose of the study, the type of tasks they would perform, and how their data would be used. (2) All data collected from participants were anonymized to protect their privacy. (3) All participants were fairly compensated for their time and contribution based on the market price. (4) All participants were given full autonomy to exit the experiments at any time without any penalty.

**Potential Risks**    We recognize that a model designed to reason about social situations could be misused or generate harmful advice if deployed improperly. To mitigate this risk, we state clearly that our work is foundational research. The CogFlow model is not intended to be a substitute for professional human judgment, nor is it designed for therapeutic, crisis intervention, or high-stakes social decision-making applications. Our goal is to enhance the transparency and interpretability of LLM reasoning in social situations, not to automate social interaction.

REPRODUCIBILITY STATEMENT

To ensure the reproducibility of our findings, we will release our full implementation of the CogFlow framework, which includes full data and code for data collection, SFT, and RL. The prompts used for data generation are provided in Appendix C. Crucial hyperparameters for training our models and the baselines are documented in Appendix D.2. The complete source code and model checkpoints will be made publicly available upon publication. We provide a temporary anonymized git repository at `https://anonymous.4open.science/r/CogFlow2025`.

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

## A USE OF LLMS

During paper writing, we used LLMs as an assistive tool to enhance the quality of the presentation. We employed LLMs to provide suggestions for grammatical corrections and polishing of the

manuscript. The core ideas, scientific arguments, and the overall structure of the paper were developed exclusively by the authors. All suggestions generated by LLMs were carefully reviewed, edited, and approved by the authors to ensure they accurately reflect our meaning.

The authors take full responsibility for all content presented in this paper, including any parts that were refined with the assistance of an LLM.

## B  GROUP RELATIVE POLICY OPTIMIZATION (GRPO)

We adopt the GRPO (Shao et al., 2024) algorithm to optimize $\pi_\theta$ with respect our reward function $\mathcal{R}$. Let $\pi_{\theta_{old}}$ denote the behavior policy from the previous iteration. For each input $x$, GRPO samples a group of cognitive flows $G = \{o_1, o_2, \cdots, o_N\}$, where each flow $o_i = (\tau_i, y_i)$. It then computes a relative advantage for each flow by normalization its reward:

$$A_i = \frac{\mathcal{R}(o_i|x) - \underset{o_j \in G}{\mathrm{mean}}(\mathcal{R}(o_j|x))}{\underset{o_j \in G}{\mathrm{std}}(\mathcal{R}(o_j|x))} \tag{7}$$

GRPO then optimizes the following objective (denote $p_{i,j} = \frac{\pi_\theta(o_{i,j}|x,o_{i,<j})}{\pi_{\theta_{old}}(o_{i,j}|x,o_{i,<j})}$):

$$\mathbb{E}_{G \sim \pi_{\theta_{old}}} \left[ \frac{1}{N} \sum_{i=1}^{N} \frac{1}{|o_i|} \sum_{j=1}^{|o_i|} \left( \min\left(p_{i,j} A_i, \mathrm{clip}_\epsilon\left(p_{i,j}\right) A_i\right) - \beta \mathrm{D}_{KL}[\pi_\theta || \pi_{ref}] \right) \right] \tag{8}$$

## C  PROMPTS FOR METHODOLOGY

### C.1  PROMPTS FOR SEED DATA COLLECTION

We used R1 (Guo et al., 2025) to generate the seed data. Each data instance consists of three components: a situation (describing the background and story), a question (based on the situation), and format constraints (specifying the required output format). First, we used the Prompt 1 to generate the situation and question. Next, we used the Prompt 2 to generate the format constraint. Finally, we used the Prompt 3 to validate the quality of the resulting data instance. If it met our quality standards, it was stored; otherwise, it was discarded, and the process was repeated from the beginning.

```
## **[Task]**

Given a scenario description and suggestions related to the scenario, you
    are required to generate a scenario and a question for the COGNITION
    TEST. You should just use the description and suggestions as
    triggers; you can convert the scenario arbitrarily by yourself:

1. **Summarize the Scenario**:
    - Objective: Craft a scene of dynamic social interaction focusing on
    several with motion-driven engagement. Describe the scenario using
    plain words.
    - It should focus on social interactions, with enough details, for
    example:
        - Environmental Context: Describe a specific time/place.
        - Specific Task: Clearly state efficient information. For example
    , the problem they are facing or the activity they are doing.
        - Character Relationship: Clearly state the relationship between
    the roles.
        - Character Dynamics: Establish clear profiles of the characters.
    - IMPORTANT: The scenario should be concise with enough details (not
    necessarily related to the original scenario or the question). It's
    better to include either relevant or irrelevant details to the
    question stated in the next step.

2. **State the question**:
```

```
15        - Concisely state the question based on the scenario in one sentence
          using the third person perspective. But you should only state the
          question simply, using words of mouth.
16        - You should double-check that the answer is NOT stated in the
          scenario. There should also be no direct/indirect hints in the
          scenario.
17        - The question should be suitable for a cognitive test. You should
          ignore the original question stated in [Scenario Description].
18
19   3. **Output Format**:
20        Present your result in JSON format using the following structure and
          respond in English. You should only use simpler vocabulary at a high
          school level to form your answers. Make sure that quotes inside all
          strings are escaped with backslashes:
21   ```json
22   {{
23        "scenario": "Scenario Summary",
24        "question": "Question in one sentence"
25   }}
26   ```
27
28   ## **#Possible Tests#**
29   These examples are way too brief and easy; your output should be more
          detailed and harder. For example, you should not give any hints, and
          it had better be open-ended.
30   ```json
31   {examples}
32   ```
33
34   ## **[Scenario Description]**
35   {scenario_description}
36
37   ## **[Suggestion]**
38   {suggestion}
```

Prompt 1: The prompt template for social situation and question generation. {examples}, {suggestion}, {scenario_description} are placeholders. {examples} is the examples randomly sampled from a manually crafted set, {suggestion} are the comments of the Reddit post, {scenario_description} is the original Reddit post.

```
1    ## **[Task]**
2
3    Given a [User Input] containing a Story and an open-ended Question,
         please propose an appropriate constraint on the output format for the
          answer to the Question containing all constraints in [Required
         Constraints]. The proposed constraint should be concise.
4
5    Note:
6    - The generated instructions cannot contain any content related to or
         hinting at the answer.
7    - The output format should follow [Output Format].
8
9    ## **[User Input]**
10   {user_input}
11
12   ## **[Required Constraints]**
13   {required_constraints}
14
15   ## **[Output Format]**
16   You should directly output the instruction as natural sentences without
         any additional words or explanations (especially explain how you
         generate the output). In one word, your whole output can be directly
         used as a constraint.
```

Prompt 2: Prompt template for constraint generation. {user_input} is the generated situation and question. {required_constraints} is randomly sampled from the options in Table 3.

Table 3: Prompts for generating output constraints.

| Constraint Type | Prompt Template Options |
| --- | --- |
| **Format** | The output should be formatted in JSON / YAML / Markdown/ Bullet or any other suitable format. To state this constraint, you should choose one specific format and give a brief demonstration of the format of the output to make it clear. Note that your instructions should be concise. **IMPORTANT: You should make sure that your demonstration is just formal, without any hint of the real answer. ** |
| **Verbosity** | • **High:** The output should be of high verbosity, which means detailed (but still needs to be concise). 
 • **Medium:** The output should be of medium verbosity, which means balancing between brief and detailed. 
 • **Low:** The output should be of low verbosity, which means brief and concise. |

```
## **[Task]**

Please check the scenario, question, and constraint given in the [User
    Input], determine in order whether the following conditions are met,
    and provide your response following the output requirements in [Check
    Output Format].

1. Please check if the question is relevant to the scenario. For example,
    the content involved must be mentioned in the scenario.
2. Please check if the constraint does not imply the answer to the
    question, but only provides formatting content or restates the
    content of the question.
3. Please check if the constraint does not contain confusing content. The
    constraint must be a reasonable format restriction for someone
    answering the question. For example, the act of "requiring in the
    constraint not to imply the answer" does not meet the requirement.

## **[User Input]**

{user_input}

## **[Check Output Format]**

Please use JSON format for the output, with only one key named 'result',
    and the value being a boolean type. true indicates that all
    requirements are met, and false indicates that at least one
    requirement is not met. Please follow the structure below:

```json
{{
        "result": true / false
}}
```
```

Prompt 3: The prompt template for seed data quality validation. {user_input} is a placeholder for the social situation, question, and format constraints.

## C.2 PROMPTS FOR COGNITIVE FLOW SIMULATION

We used R1 to perform the following tasks:

- **Planning**: LLMs are used in the following two operations: **Generation**: To generate a cognitive unit's thought content, we use the Prompt 4. **Prediction**: The Prompt 5 to identify the most relevant subsequent units. The choice `Terminate` stands for the reasoning has reached a terminal state.

- **Completion**: We prompted the LLMs using the Prompt 6 to get the final response.

```
## **[Task]**
- Background: You are an assistant helping to answer problems. You need
   to carry out the next step [{node_name}] of a reasoning chain to help
    respond to [User Input].
- Requirements:
   * follow the instructions in **Reasoning Step: [{node_name}]** which
    defines the reasoning step.
   * It should be the next step of the half-finished reasoning chain in [
    Existing Analysis]. The result can only be not aligned with the
    analysis in [Existing Analysis] if you need to fix mistakes or
    explore aspects not considered. You can refer to [Analysis
    Expectation] for guidance, but you do not need to strictly follow it.
   * **Important**: Your output should be specific, without fake
    information.
   * **Important**: Your output should be comprehensive, including any
    possible aspects.
   * **Important**: Your output should only contain one step. If other
    things are in need, state the need and reserve the reasoning for the
    next steps.
   * You should only use simpler vocabulary at a high school level to
    form your answers.

## **[User Input]**
{user_input}

## **[Existing Analysis]**
{previous_nodes}

## **[Analysis Expectation]**
{analyze_expect}

## **Reasoning Step: [{node_name}]**
{node_description}

## **[Output Format]**
Please output in English. The content should be a smooth and coherent
    paragraph, following the format below:
```json
{{
        "content": "the content of the required step"
}}
```
```

Prompt 4: The prompt template for generating cognitive units in the cognitive flow simulation. {user_input}, {previous_nodes}, {analyze_expect}, {node_description}, {node_name} are placeholders. The {node_name} and {node_description} are prompts from Table 4. The {analyze_expect} is the justification for choosing this unit, which was generated during the unit selection.

```
## **[Task]**
- Background: You are an assistant helping with problems. You need to
   choose the next step of a reasoning chain to help respond to the user
   's input in [User Input]. The chain should be comprehensive.
- Requirements:
   * You should select **ALL** possible candidates from [Candidate Next
    Steps] that can be a reasonable next ONE step of the half-finished
    reasoning chain provided in [Existing Analysis]. You could visit the
    same step several times to get more information or analyze further.
```

Table 4: Prompt templates for cognitive units.

| Prompt Type | Prompt Template |
|---|---|
| Observation | - **Task**: Observe and interpret the specific behaviors, attitudes, or other information from the current context. The extracted facts must be precise and detailed without vague information. **NOTE: ALL THE INFORMATION MUST BE ALIGNED WITH THE CONTEXT. DO NOT MAKE UP FAKE INFORMATION.** 
 - **Output**: State observation comprehensively. |
| Attribution | - **Task**: Attribute and evaluate the events or behaviors. It might include: Causal reasoning for others' actions / Impact assessment on current context / Further analysis and explanation. 
 - **Output**: State the specific reason comprehensively. |
| Motivation | - **Task**: Generate motivation and goals, addressing the main problem discovered in other steps. 
 - **Output**: State the goal and motivation. |
| Regulation | - **Task**: Check and adjust the previous thought to form a revised motivation or perception, or action plan. You should check (1) whether it lacks consideration, (2) whether other requirements need to be noticed. Think of the effect of the current plan or behavior, and check if there exists any risk. You should also check if there are any misunderstandings and be suspicious of the information in the analysis. 
 - **Output**: Accurately and comprehensively state the problem and how to solve it. |
| Efficacy | - **Task**: Assess the internal perceptions, emotions, and beliefs of the actor of some behavior, and adjust the perception or action plan. 
 - **Output**: State the efficacy and adjustment of action. |
| Behavior | - **Task**: Determine a more complete behavior based on the current environment and the analysis. |

```
 5    * If analysis is sufficient for responding to the [User Input], and
      there are no concerns, DIRECTLY select the [Terminate] step. (NOTE:
      If you are not certain or you think there might be other potentials,
      you must choose other nodes along with Terminate. )
 6    * If you find some bad steps in [Existing Analysis] (for example:
      misinformation, unclear statement, etc. ), redoing it again might
      refine it.
 7    * If more than one valid options exist, list the most applicable 2 or
      3 steps, and put the most applicable one in the first place.
 8    * The names of the next steps should be exactly the same as the name,
      e.g., Attribution and Evaluation.
 9    * You should first review the prior steps in [Existing Analysis], and
      then determine the candidates for the next step.
10
11
12 ## **[User Input]**
13 {user_input}
14
15 ## **[Existing Analysis]**
16 {previous_nodes}
17
18 ## **[Candidate Next Steps]**
19
20 - **[Observation]**
21    * Observe the specific behaviors or attitudes from the current context
      .
22
23 - **[Regulation]**
```

```
24      * Validate and refine previous thoughts: (1) consider twice to polish
         the thought, behavior, or motivation, (2) check if there exists more
         information in the scenario that needs to be considered.
25
26   - **[Behavior]**
27      * derive context-specific behaviors.
28
29   - **[Efficacy]**
30      * analyze and adjust internal perceptions of the scene and action plan
         .
31
32   - **[Attribution]**
33      * further interprets the result of previous steps, may include Causal
         reasoning for others' actions, or Impact assessment on the current
         context.
34
35   - **[Terminate]**
36      * Terminate analysis, synthesize final conclusion, and respond to the
         user.
37
38   - **[Motivation]**
39      * formulate one's primary drivers of oneself, based on their needs/
         desires identified in other steps.
40
41   ## **[Output Format]**
42   ```json
43   {{
44       "rationale": "Concise justification for selecting the next one step
         candidates, and choose the most likely one",
45       "next_step_candidates": ["step name", ...]
46   }}
47   ```
```

Prompt 5: The prompt template for choosing the next cognitive units in the cognitive flow simulation. {user_input}, {previous_nodes} are placeholders. {user_input} is the social situation, {previous_units} is a linear chain of existing cognitive units. LLMs are required to predict the units that directly follow the end of the sequence.

```
1   {user_input}
2   Please answer under the guidance of the following thought:
3   <|begin think|>
4   {previous_units}
5   <|end think|>
```

Prompt 6: The prompt template for generating a response under the guidance of cognitive flow. {user_input}, {previous_units} are placeholders. {user_input} is the social situation, {previous_units} is the simulated cognitive flow.

### C.3 PROMPTS FOR DUAL-VALIDATION BASED FILTERING

We used R1 to perform the following tasks:

- **Comparison Pool Construction**: We randomly gathered snippets from the generated cognitive flows, and reused the Prompt 6 to get the final response under the guidance of the reconstructed fake cognitive flow.
- **Two-stage Comparative Preference Ranking**: For both two stages in preference ranking, we use a unified Prompt 7, which can score the first response listed to be 5 as an anchor.

```
1   **[Task]**
2
3   Given the [User Input] and the corresponding multiple answers in [Answers
     ] (which are in random order), please score these answers on a scale
     of 1-10 (the higher the score, the better) according to the following
     principles:
```

```
4
5  (1) Based on the [User Input] and all the answers in [Answers], propose
        evaluation criteria that can assess the quality of the given answers.
         Ensure that under these criteria, the first answer listed receives a
         score of 5.
6  (2) Explain the scoring principle for each score value sequentially.
7  (3) Score all the answers in [Answers] based on the established
        evaluation criteria. You must use the first listed answer as a 5-
        point reference sample and provide a reason for each score.
8  (4) Refer to the [Output Format] for the output structure.
9
10 Note: You must ensure that the scores for the [Answers] are well-
        differentiated.
11
12 Special Attention: You must ensure that the first answer listed receives
        a score of 5 under your scoring standard. Use this first answer as a
        baseline (referred to as "Baseline"). For subsequent answers, a score
         greater than 5 must mean it is better than the first answer, and a
        score less than 5 must mean it is worse than the first answer.
13
14 **[User Input]**
15 {user_input}
16
17 **[Answers]**
18 {answers}
19
20 **[Output Format]**
21 Output in JSON format, with every answer giving one score in 1-10. The
        answer is identified by 'id'.
22 ```json
23 {{
24     "think": "analyze the user's input, come up with some criterion",
25     "standard": [
26         {{
27             "score": 10,
28             "standard": "standard of score 10"
29         }},
30         ...
31         {{
32             "score": 5,
33             "standard": "standard of score 5"
34         }},
35         ...
36         {{
37             "score": 1,
38             "standard": "standard of score 1"
39         }},
40     ],
41     "result": [
42         {{
43             "id": ...,
44             "reason": "compare with the first answer (Baseline), and then
        judge the quality of this answer",
45             "score": evaluated socre
46         }},
47         ...
48         {{
49             "id": ...,
50             "reason": "compare with the first answer (Baseline), and then
        judge the quality of this answer",
51             "score": evaluated score
52         }}
53     ]
54 }}
55 ```
```

Prompt 7: The prompt template for Two-Stage Comparative Preference Ranking. {user_input}, {answers} are placeholders. {answers} is a list of answers to be evaluated, each with a unique integer id. {user_input} is the social situation with question and format constraint.

## C.4 PROMPTS FOR COGNITIVE FLOW PRUNING

LLMs were used to evaluate the quality of cognitive flows with Prompt 8. We screened out those who scored below 4 in at least one category.

```
**[Task]**

Please evaluate the cognitive flow provided in the [Reasoning Flow] based
    on the three core criteria listed below. You need to score each
    criterion independently on a scale of 1-10 (the higher the score, the
    better) and provide a reason for each score.

Evaluation Criteria:
- **Coherence**: Is it logically sound and free of internal
    contradictions?
- **Interpretability**: Does it clearly explain the social dynamics or
    core mechanisms involved?
- **Predictability**: Does it offer reasonable insight into the future
    evolution of the social dynamics?

Please strictly follow the JSON format required in the [Output Format].

**[Reasoning Flow]**
{reasoning_flow}

**[Output Format]**
Please output in JSON format. The JSON structure should include your
    thought process, the independent scores, and reasons for each
    criterion.
```json
{{
    "think": "evaluation process",
    "evaluation_result": {{
        "coherence": {{
            "reason": "Explain your reasoning",
            "score": evaluated_score
        }},
        "interpretability": {{
            "reason": "Explain your reasoning",
            "score": evaluated_score
        }},
        "predictability": {{
            "reason": "Explain your reasoning",
            "score": evaluated_score
        }}
    }}
}}
```
```

Prompt 8: The prompt template for cognitive flow evaluation. {reasoning_flow} is a placeholder for the cognitive flow to be evaluated.

Table 5: Distribution of social situations. Percentages represent the proportion of the total dataset.

| Category / Subcategory | Count | Percentage |
|---|---|---|
| **Romance** | **819** | **16.06%** |
|   Dating & Courtship | 108 | 2.12% |
|   Romantic Challenges | 620 | 12.16% |
|   Long-term Partnership | 91 | 1.78% |
| **Family** | **1,414** | **27.73%** |
|   Parent-Child Interaction | 351 | 6.88% |
|   Major Family Events & Issues | 663 | 13.00% |
|   Extended Family Relations | 295 | 5.78% |
|   Household & Logistics | 105 | 2.06% |
| **Public** | **690** | **13.53%** |
|   Stranger Encounters | 184 | 3.61% |
|   Community Life | 422 | 8.27% |
|   Service Interactions | 84 | 1.65% |
| **Friendship** | **1,265** | **24.80%** |
|   Intimate Friendship | 723 | 14.18% |
|   Group Activities & Events | 383 | 7.51% |
|   Casual Hangouts | 159 | 3.12% |
| **Professional** | **912** | **17.88%** |
|   Professional/Academic Challenges | 396 | 7.76% |
|   Professional Relationships | 274 | 5.37% |
|   Task-Oriented Collaboration | 242 | 4.75% |
| **Total** | **5,100** | **100.00%** |

# D EXPERIMENTS

## D.1 MORE DETAILS OF OUR DATASET

**Detailed Information of Reddit Data**  We use anonymized Reddit[1] posts as our seed situations. The subreddits we used are as follows: FriendshipAdvice, LifeAdvice, Advice, AskWomenOver30, emotionalsupport, family, relationship_advice, confessions, socialskills, AmItheAsshole, AskMenOver30, AskMen, DecidingToBeBetter, mentalhealth, Anxiety, AskWomen, SocialEngineering, and familyadvice.

**Distribution of Social Situations**  The distribution of social situation categories in our dataset is listed in Table 5.

**Distribution of Test Set Difficulty**  Following experts' annotations, the test set instances were classified into three levels of difficulty: Easy (137), Medium (232), and Hard (131).

## D.2 IMPLEMENTATION DETAILS OF OUR MODELS AND BASELINES

**Experimental Setup**  All experiments were conducted on 8x NVIDIA H20 GPUs, using Llama-3.1-8B-Instruct (Meta, 2024) and Qwen-2.5-7B-Instruct (Yang et al., 2024) as base models. For the training pipeline, we employed the LLaMA-Factory framework[2] (Zheng et al., 2024) for the SFT and the veRL[3](Sheng et al., 2024) engine for RL.

**Training Hyperparameters**  For the **SFT** stage, the model was trained for 2 epochs with a batch size of 8, a learning rate of $5 \times 10^{-5}$, and a context length of 8,192. The **preference reward**

---

[1]https://www.reddit.com

[2]https://github.com/hiyouga/LLaMA-Factory

[3]https://github.com/volcengine/verl

**model**, based on Qwen-2.5-7B-Instruct, was augmented with a linear classifier head to predict reward scores. It was trained for 1 epoch with a batch size of 8 and other hyperparameter settings the same as SFT. During the **RL** stage, we generated trajectories by performing 6 rollouts for each of the 24 social situations in a training batch. The policy was subsequently updated using a mini-batch size of 4 situations, resulting in 6 gradient updates per collection batch. We use a learning rate of $10^{-6}$, and a KL divergence coefficient to $10^{-3}$. By default, we set $\omega_1 = 1, \omega_2 = 0.05$, and $\omega_3 = 0.1$.

**CogFlow Models**   Our primary models are fine-tuned from the base model using cognitive flow.
- **CogFlow-SFT**: The base model fine-tuned on our SFT dataset.
- **CogFlow**: The final model, fine-tuned from CogFlow-SFT using our complete RL reward function.
- **CogFlow-GRPO**: An ablation of our method, fine-tuned from CogFlow-SFT using only the response quality reward ($\omega_1 = 1, \omega_2 = 0, \omega_3 = 0$). This is equivalent to the GRPO algorithm.
- **CogFlow (w/o $\mathcal{R}_{\mathrm{Div}}$)**: An ablation fine-tuned from CogFlow-SFT, excluding the diversity reward component ($\omega_1 = 1, \omega_2 = 0, \omega_3 = 0.1$).
- **CogFlow (w/o $\mathcal{R}_{\mathrm{Len}}$)**: An ablation fine-tuned from CogFlow-SFT, excluding the length penalty component ($\omega_1 = 1, \omega_2 = 0.05, \omega_3 = 0$).

**Tuning-free Models**   For models that do not have a native long chain-of-thought ability, such as GPT-4o and DeepSeek-V3, we employed a zero-shot Chain-of-Thought (CoT) prompting strategy to elicit step-by-step reasoning. Specifically, we appended the following instruction to the end of each input prompt: `Let's think step by step, and use <FINAL RESPONSE> before you give the final answer.`

**'Direct-' Models**   These models are trained to generate the final response directly, without any explicit reasoning process.
- **Direct-SFT**: It is fine-tuned from the base model using the CogFlow SFT dataset, but with the reasoning process removed.
- **Direct-GRPO**: Fine-tuned from Direct-SFT using GRPO. It only used response quality reward $\mathcal{R}_{\mathrm{Res}}$.

**'Distilled-R1-' Models**   These models are designed to emulate the R1-style reasoning format, effectively serving as distilled versions of R1.
- **Distilled-R1-SFT**: Fine-tuned from the base model using all the social situations of CogFlow's SFT data, but DeepSeek-R1 directly generates the reasonings and responses.
- **Distilled-R1-GRPO**: Fine-tuned from Distilled-R1-SFT using GRPO. The reward function combines response quality with a format-checking reward, $\mathcal{R}'_{\mathrm{Format}}$, which verifies the presence of `<think>` and `</think>` tags. The total reward is:

$$\mathcal{R} = \mathcal{R}'_{\mathrm{Format}} \cdot \mathcal{R}_{\mathrm{Res}} \tag{9}$$

- **Distilled-R1-GRPO$_{\mathcal{R}_{\mathrm{Len}}}$**: This configuration is identical to Distilled-R1-GRPO but incorporates our reasoning length reward, $\mathcal{R}_{\mathrm{Len}}$, to encourage more concise reasoning paths. The total reward is:

$$\mathcal{R} = \mathcal{R}'_{\mathrm{Format}} \cdot (\omega_1 \cdot \mathcal{R}_{\mathrm{Res}} + \omega_3 \cdot \mathcal{R}_{\mathrm{Len}}) \tag{10}$$

### D.3 IMPLEMENTATION DETAILS OF LLM EVALUATORS

We detail the LLM-based and reward model-based evaluators referenced in Table 1, Table 2, and Table 6 below:

**Prompt-Based Direct Scoring Evaluators**   These evaluators generate a direct score for each response individually. They use R1 (Guo et al., 2025) (Score-R1) and Qwen3-32B (Yang et al., 2025) (Score-Q32B) as the evaluators, both prompted with the template from Prompt 9.

```
**[Task]**

```

```
 4  Given a [User Input] and its corresponding [Answer], please provide a
        comprehensive score between 1 and 10 based on the quality of the
        answer, where a higher score indicates a better answer. A score of 5
        indicates that the answer is basically correct but may be incomplete,
        unclear, or partially inaccurate.
 5
 6  Scoring Criteria Explanation (for reference; please make a comprehensive
        judgment):
 7
 8  - 10: Perfect answer. Entirely accurate, informative, well-structured,
        and appropriately worded. Effectively addresses the user's query,
        potentially even exceeding expectations.
 9  - 8-9: Excellent answer. Accurate and complete in information, logically
        clear, fluently expressed, fully satisfying the user's needs.
10  - 6-7: Good answer. Basically correct and relevant, but may lack depth in
        certain details or contain minor inaccuracies.
11  - 5: Passable answer. Generally correct but potentially incomplete,
        somewhat unclear, or containing individual errors that do not
        severely impact understanding.
12  - 3-4: Insufficient answer. Partially relevant but missing key
        information, containing significant errors, or failing to address the
        core issue.
13  - 1-2: Poor answer. Severely off-topic, containing incorrect information,
        or entirely unhelpful.
14
15  When evaluating, you may comprehensively consider the following
        dimensions (not all are required):
16  - Accuracy: Whether the answer is factually correct and non-misleading.
17  - Completeness: Whether it covers the key points of the user's question.
18  - Relevance: Whether the answer stays closely aligned with the user's
        question without deviating from the topic.
19  - Clarity: Whether the expression is clear, easy to understand, and well-
        organized.
20  - Practicality: Whether it offers practical help to the user and is
        actionable (if applicable).
21
22  **[User Input]**
23  {user_input}
24
25  **[Answer]**
26  {answer}
27
28  **[Output Format]**
29  Output in JSON format, with the answer given one integer score in 1-10.
30  ```json
31  {{
32      "score": evaluated score
33  }}
34  ```
```

Prompt 9: The prompt template for direct scoring response. {user_input}, {answer} are placeholders. {answer} is the answer to be evaluated. {user_input} is the social situation with question and format constraint.

**Comparative Preference Ranking Evaluators**   These evaluators generate scores for a batch of responses simultaneously using comparative ranking methods. We apply two distinct methodologies to both R1(Guo et al., 2025) and Qwen3-32B(Yang et al., 2025) models:

- **CPRank**: This is a direct comparison method. The evaluator is prompted (using Prompt 7) to rank all responses within a given batch from best to worst in a single pass, thereby establishing a complete preference order at once. It results in CPRank-R1 and CPRank-Q32B.
- **CPRank$^2$**: This method, as described in the main text, involves two steps to refine the evaluation. First, for **initial ranking**, the model generates situation-specific criteria and uses them to assign an initial score and critique for each response. Second, for **comparative reranking**, it selects the

Table 6: Results of automatic evaluation for more tuning-free models with CoT strategy.

| Models | CPRank$^2$-R1 ($\uparrow$) | | | | CPRank$^2$-Q32B ($\uparrow$) | | | | Reasoning Length (tokens, $\downarrow$) |
|---|---|---|---|---|---|---|---|---|---|
| | Overall | Easy | Medium | Hard | Overall | Easy | Medium | Hard | |
| **Tuning-free Models** | | | | | | | | | |
| Llama-3.1-70B (CoT) | 0.0619 | 0.1016 | 0.0577 | 0.0264 | 0.1256 | 0.2016 | 0.1173 | 0.0607 | 231.33 |
| Qwen-2.5-7B (CoT) | 0.0820 | 0.1208 | 0.0849 | 0.0403 | 0.1340 | 0.2178 | 0.1255 | 0.0612 | 184.90 |
| Llama-3.1-8B (CoT) | 0.0885 | 0.1000 | 0.0882 | 0.0772 | 0.1679 | 0.2467 | 0.1575 | 0.1039 | 253.12 |
| Qwen-2.5-72B (CoT) | 0.1401 | 0.2010 | 0.1703 | 0.0465 | 0.1487 | 0.2588 | 0.1338 | 0.0598 | 242.75 |
| DeepSeek-V3 (CoT) | 0.2443 | 0.3077 | 0.2390 | 0.1862 | 0.3591 | 0.3854 | 0.3662 | 0.3192 | 927.40 |
| GPT-4o (CoT) | 0.2542 | 0.3366 | 0.2407 | 0.1876 | 0.3489 | 0.3861 | 0.3521 | 0.3035 | 918.55 |

Table 7: Results of pairwise comparison. The three numbers are the percentage of *win/tie/loss* for the paired models.

| Models | Compared Models | Easy (%) | Medium (%) | Hard (%) | Overall (%) |
|---|---|---|---|---|---|
| CogFlow vs. | Simulated-CogFlow | 56.9 / 0.8 / 42.3 | 43.1 / 0.9 / 55.9 | 50.6 / 1.9 / 47.4 | 49.1 / 1.2 / 49.7 |
| | DeepSeek-R1 | 55.5 / 0.0 / 44.5 | 49.7 / 1.0 / 49.2 | 52.2 / 1.6 / 46.2 | 51.9 / 1.0 / 47.0 |
| | Distilled-R1-GRPO$_{\mathcal{R}_{\text{Len}}}$ | 64.6 / 4.4 / 31.0 | 54.7 / 3.0 / 42.3 | 62.6 / 3.7 / 33.7 | 59.9 / 3.6 / 36.5 |
| Simulated-CogFlow vs. | DeepSeek-R1 | 47.1 / 4.3 / 48.6 | 55.2 / 1.7 / 43.1 | 54.9 / 4.7 / 40.4 | 52.9 / 3.6 / 43.6 |
| | Distilled-R1-GRPO$_{\mathcal{R}_{\text{Len}}}$ | 61.1 / 1.6 / 37.3 | 56.1 / 1.7 / 42.2 | 54.8 / 1.3 / 43.9 | 56.9 / 1.6 / 41.5 |
| DeepSeek-R1 vs. | Distilled-R1-GRPO$_{\mathcal{R}_{\text{Len}}}$ | 54.0 / 0.7 / 45.3 | 52.2 / 0.0 / 47.8 | 49.2 / 1.6 / 49.2 | 51.6 / 0.8 / 47.6 |

median-ranked response as an anchor to mitigate scoring biases (e.g., positional bias). The model then performs a final comparative reranking of the entire pool against this anchor, yielding a more robust preference order. It results in CPRank$^2$-R1 and CPRank$^2$-Q32B.

- **RM$_\phi$**: This evaluator is the reward model specifically trained to score responses described in 3.3. The implementation details are described in D.2.

## D.4 Detailed Information of Models Used in Human Evaluation

We detail the models mentioned in the pairwise comparison here:
- **CogFlow**: Llama-3.1-8B-Instruct fine-tuned using the whole CogFlow pipeline.
- **Distilled-R1-GRPO$_{\mathcal{R}_{\text{Len}}}$**: Llama-3.1-8B-Instruct fine-tuned using the Distilled-R1-GRPO$_{\mathcal{R}_{\text{Len}}}$ method.
- **Simulated-CogFlow**: The pruned results of cognitive flow simulation (crafted by prompting R1).
- **DeepSeek-R1**: The native DeepSeek-R1 model.

## D.5 More Baseline Performance

More baseline results (GPT-4o, DeepSeek-V3, Qwen-2.5-7B/72B-Instruct and LLama-3.1-8B/70B-Instruct using CoT strategy stated in D.2) are shown in Table 6.

## D.6 Performance of Pairwise Comparison

We show the precise pairwise results from the experts' pairwise evaluation in Table 7.

To quantitatively assess the relative strength of our models from pairwise comparison data, we employ the Bradley-Terry (BT) model(Bradley & Terry, 1952), a statistical method for converting pairwise preferences into a continuous capability scale. The results are provided in Figure 3.

**Objective Function** The core assumption of the BT model is that each model $i$ possesses an unobserved strength parameter $p_i \in \mathbb{R}_{>0}$. The probability of model $i$ winning against model $j$ is given by:

$$P(i \text{ defeats } j) = \frac{p_i}{p_i + p_j}. \tag{11}$$

Given the observed number of model $i$ defeats model $j$ $W_{ij}$, our objective is to find the set of strength parameters $p = \{p_1, p_2, ..., p_n\}$ that maximizes the log-likelihood of the observed outcomes. The

total log-likelihood function is:

$$\mathcal{L}(\mathbf{p}) = \sum_{i=1}^{n} \sum_{j \neq i}^{n} W_{ij} \log \left( \frac{p_i}{p_i + p_j} \right) \tag{12}$$

$$= \sum_{i=1}^{n} \left( \left( \sum_{j \neq i} W_{ij} \right) \log(p_i) - \sum_{j \neq i} W_{ij} \log(p_i + p_j) \right). \tag{13}$$

**Optimization Method** Since a closed-form solution for maximizing this likelihood is not available, we use an iterative algorithm to find the Maximum Likelihood Estimates (MLE) for the parameters p. The update rule for each parameter $p_i$ at each iteration is derived from the likelihood equations, resulting in the following fixed-point iteration scheme:

$$p_i^{(\text{new})} = \frac{\sum_{j \neq i} W_{ij}}{\sum_{j \neq i} \frac{W_{ij} + W_{ji}}{p_i^{(\text{old})} + p_j^{(\text{old})}}}. \tag{14}$$

The proof of its convergence and optimality is detailed in (Hunter, 2004). Here, we initialize all $p_i$ to 1 and then perform iterative updates. Each update step first applies Eq. 14 for $i = 1, ..., n$, and then normalizes the resulting parameter vector $\left( p_1^{(\text{new})}, ..., p_n^{(\text{new})} \right)$. This process is repeated until the parameters converge, defined as when the $L_2$ norm of the parameter vector change is below a tolerance of $10^{-6}$. Finally, to ensure a unique solution, the parameters are normalized such that the weakest model has a score of 1.

### D.7 PROMPTS AND EXAMPLES FOR COGNITIVE INTERVENTION FOR HUMANS

We use R1 to translate reasoning chains into natural language interventions, following the Prompt 10. To illustrate this, Table 8 presents two intervention examples derived from two reasoning styles.

```
## **[Task]**
You are a thoughtful and persuasive mentor. Your friend encountered a
    task: [Question]
He has been provided with several responses, the best one is [
    best_response], and the rest are [other_responses].
But he did not choose the best one.
Now, you plan to persuade his friend to reconsider. But you should be
    gentle, so you should take a reasoning procedure [Best Reasoning]
    leading to the best response, and try to (1) figure out based on the
    chosen response, what may not he considered in each step of the
    reasoning procedure, and (2) try to teach him to think in better ways
    . You can guide him to build up the reasoning procedure(You should
    assume that he is a beginner, and he may not know the reasoning
    procedure. ), and make every node of the reasoning procedure better.

## **[Question]**
{question}

## **[other_responses]**
{other_responses}

## **[best_response]**
{best_response}

## **[Best Reasoning]**
{best_reasoning}

## **[Requirement]**
- YOU SHOULD NEVER MENTION OR HINT AT THE EXISTENCE OF THE [best_response
    ].

## **[Output format]**
```

```
23
24 You should persuade him to reconsider. The persuasion should be concise.
     Please provide the persuasion in JSON format like this:
25 ```json
26 {{
27     "words": "Your persuasion"
28 }}
29 ```
```

Prompt 10: Prompt converting reasoning to intervention. {question}, {other_responses}, {best_responses}, {best_reasoning} are placeholders. {other_responses} is a list of all four candidate responses from CogFlow, Simulated-CogFlow, DeepSeek-R1, and Distilled-R1-GRPO$_{\mathcal{R}_{\text{Len}}}$. {question} is a social situation and question. {best_responses} and {best_reasoning} are the response content and reasoning of the ground truth model in the situation, respectively.

Table 8: Example of guidance for intervention for two styles of reasoning, i.e., cognitive reasoning vs. R1 reasoning.

| Type | Guidance |
| --- | --- |
| **Cognitive Flow Style** | Hi there! I noticed you analyzing that school scenario - great focus on the visible emotions. Let's build your analysis step by step together. First, we should collect every observable fact neutrally, like Leo would: Maya's bracelet-twisting, the folded note exchange, Chris's smirk-to-whisper sequence, her abrupt step back and near-trip, the library retreat, Chris's chuckle, and the bell ringing. Now, when interpreting, we might consider adding layers: Could the note's secrecy create uncertainty about its purpose? Might Chris's immediate chuckle after her exit amplify concerns about intent? And importantly, does Maya's established anxiety before the inter-action suggest potential patterns? The time pressure from the bell could also heighten Leo's urgency. This method helps ensure we connect all details to concerns without speculation. Want to walk through how this changes your interpretation? |
| **R1 Reaoning Chain Style** | Hey, let's walk through this step-by-step together. First, consider Maya's nervousness before the interaction - her bracelet-twisting, darting eyes, and untied shoelace show she's already anxious. Many beginners might overlook how these details establish vulner-ability. Next, Chris's smirk while reading the note and his whis-pered comment are crucial - they suggest negative intent rather than kindness. Maya's physical reaction of stepping back abruptly and nearly tripping reveals shock or fear, directly linking to what Chris said. Then note her hasty retreat to the library (a safe space) instead of staying social, showing deep distress. Chris's solitary chuckle afterward confirms he wasn't being supportive. Finally, Leo's stopped eating and wrinkled forehead prove he's connecting these dots in real-time. By building the analysis this way - starting with initial cues, then reactions, then consequences - we see how every detail points to Leo worrying about emotional harm to Maya. What stands out most to you in this sequence? |

