# OpenReview forum: "Think Socially via Cognitive Reasoning"
_ICLR.cc/2026/Conference — Submitted to ICLR 2026_

### Official Review · Reviewer_4JhK · 2025-10-28

**Soundness:** 3
**Presentation:** 3
**Contribution:** 3
**Rating:** 6
**Confidence:** 3

**Summary:**

This paper proposes a training framework to improve the social reasoning capability of small LMs via model distillation. A social cognitive reasoning dataset is constructed based on initial social situation descriptions collected online and a set of heuristics using LLMs with reasoning capability. Those data are then used to improve small LMs via supervised fine-tuning and reinforcement learning with auxiliary rewards. The evaluation results indicate the proposed framework leads to better social reasoning performance compared to directly using the same technique individually (e.g. SFT, RL, distillation).

**Strengths:**

The paper is well written with a clear presentation of motivation, method, and evaluation.

The proposed training framework is grounded in social cognition theory and potentially applies to general social reasoning situations.

The evaluations are thorough with both LLM-as-judge automatic judgment and human judgment.

**Weaknesses:**

The proposed framework heavily relies on the reasoning capability of LLMs since the data is generated and augmented with LLMs based on a structured workflow with handcrafted heuristics, then evaluated all by LLMs as well. As mentioned in the paper, the robustness of this process may be influenced by the internal bias within the specific LLM. Additionally, to what degree the cognitive flow simulation applies to an out-of-distribution social situation that is not generated via the proposed pipeline is not clear. For example, situations with partial observation and long-horizon interaction where the heuristics may not generalize well.

From the AL/ML perspective, the main contribution of this work is a structured social reasoning framework to generate supervised data for model distillation and corresponding auxiliary rewards used in RL training. While the results on human decision-making are interesting, they are somewhat disconnected from the main technical contribution. Because It is less clear if the same heuristics helping humans can explain why they work for LLMs.

Table 1 needs more context to make sense. In pair-wise comparisons, an agreement ratio kappa around 0.6 seems to be relatively low. This means only ~70% of the time the LLM evaluators rank the pair consistently with humans. This leads to the concern of to what degree the LLM evaluator represents human preference in social reasoning situations.

In section 4.4, the concept of reasoning depth is used before being properly defined.

The paper appears to involve an intense human subject study but does not disclose sufficient information regarding the details and ethical approvals (e.g., IRB). For example, How were the annotators and volunteers recruited? How much were they paid and for how many hours of participation? What were their demographic backgrounds and knowledge of LLMs? Did participants know the experiment design (e.g., the source of different example instances)? Also given that the study uses online-collected content modified by LLMs, I am concerned about the safety filtering process mentioned. Could the authors please elaborate on this process?

The design decision of using majority voting in human annotation lacks rationale. As motivated in the introduction, social reasoning in many situations is by nature ambiguous and may be impacted by personal background and preference. It makes more sense to have independent judges and take the individual difference among the human population into consideration.

Presentation format may be a confounded variable in both the benchmark and two human decision-making experiments. Because the cognitive flow is generally shorter and structured with bullet points while CoT is usually a long chunk of plain text. It is difficult to distinguish whether the effect observed between conditions is due to format or content. An ablation study may be helpful.

**Questions:**

Although the introduction motivates the work to extend model training to social scenarios with ambiguous cues and no definitive outcomes, the proposed framework actually clearly defines those cues and outcomes via heuristics and powerful LLMs, then distills data to train the small model. This seems to be contradictory. Please clarify.

---

> ### Author Response · Authors · 2025-11-21
> **Response to Reviewer 4JhK (1/4)**
>
> We sincerely thank the reviewer for the constructive feedback and for recognizing the soundness of our motivation and the thoroughness of our evaluation. Below, we address the specific concerns raised in the "Weaknesses" and "Questions" to clarify our contributions and methodology.
>
> ### **Q-1: Regarding LLM bias and the generalizability of cognitive flow simulation.**
>
> **A-1:** We agree that reliance on LLMs can introduce internal bias. As a result, we implemented a rigorous approach to mitigate this risk and ensure robustness across diverse social contexts in our paper.
>
> **(1) Mitigating internal bias.** We address bias through our "dual-validation based filtering" mechanism and extensive human verification.
> - Dual-validation based filtering: We employed a two-stage comparative preference ranking specifically to mitigate scoring biases (e.g., positional bias) inherent in teacher models. Furthermore, we apply theory-grounded pruning (checking coherence, interpretability, and predictability) as a hard filter to ensure quality beyond simple preference scores.
> - Human verification: First, we confirmed that our automatic evaluator aligns with human judgment (moderate agreement in Table 1), ensuring it is a reliable proxy. Second, domain experts inspected our generated data, yielding a 96.8% pass rate, which confirms the quality of the simulation.
>
> **(2) Generalizability to OOD situations:** We improve generalizability in three ways.
> - First, our cognitive reasoning paradigm is grounded in social cognitive theory. The paradigm is explicitly designed to handle the interpretive process of analyzing ambiguous cues. This makes it naturally more robust to OOD situations, e.g., partial observation mentioned by the reviewer.
> - Second, our training data spans 16 diverse real-world categories of social situations, providing a broad foundation for learning generalizable social dynamics.
> - Third, our multi-objective RL guided by cognitive flexibility reward encourages the model to autonomously explore better reasoning paths and avoid pattern collapse (as shown in Figure 5). This RL stage improves the generalizability of models beyond the SFT dataset.
> - Finally, to empirically demonstrate this, we evaluated our CogFlow models on multiple OOD tasks, i.e., MATH-500, SocialIQA, HiToM. The detailed experimental setups are presented in the response (A-4) to the reviewer UFD7. As shown in the table below, our model maintains superior performance, validating the generalization capability of our framework.
>
> | Model                 | MATH-500      | MATH-500              | SocialIQA    | SocialIQA            | HiToM        | HiToM                |
> | --------------------- | ------------ | -------------------- | ------------ | -------------------- | ------------ | -------------------- |
> |                       | Accuracy (↑) | Reasoning Length (tokens, ↓) | Accuracy (↑) | Reasoning Length (tokens, ↓) | Accuracy (↑) | Reasoning Length (tokens, ↓) |
> | Llama-3.1-8B-Instruct | 0.4220       | -                    | 0.7070       | -                    | 0.1942       | -                    |
> | Llama-3.1-8B-Instruct (CoT)    | 0.4400       | 308.18               | 0.6760       | 249.13               | 0.3344       | 722.64               |
> | Distilled-R1-GRPO     | 0.4420       | 1034.54              | 0.6630       | 399.27               | 0.3571       | 1209.72              |
> | CogFlow (Ours)          | 0.4460       | 500.27               | 0.7042       | 227.76               | **0.3625**       | 412.99               |
> | Distilled-R1-adapt    | 0.4900       | 449.88               | 0.7573       | 348.26               | -            | -                    |
> | **CogFlow-adapt (Ours)**     | **0.5340**       | 304.86               | **0.7867**       | 166.28               | -            | -                    |

---

> ### Author Response · Authors · 2025-11-21
> **Response to Reviewer 4JhK (2/4)**
>
> ### **Q-2: The connection between the technical contribution and the human study.**
>
> **A-2:** We would like to clarify that the human decision-making study is not a disconnected finding, but rather a validation of our core premise.
>
> Our work's premise is that the conventional, logical, step-by-step reasoning paradigm is not well-suited for navigating ambiguous social situations. We propose that a paradigm modeled on human social cognition, which we term Cognitive Reasoning, should be more effective.
>
> Our technical contribution is the CogFlow framework to instill this reasoning paradigm into an LLM. The human studies tested the value of the paradigm itself, thereby validating the foundation upon which our technical work is built.
>
> Our experiments showed the parallel benefit in reasoning efficiency for both humans and LLMs:
> - For LLMs: As shown in the case study of Figure 1, conventional reasoning can fall into "cognitive rumination", producing overly long and inefficient analyses. Our CogFlow framework, guided by the cognitive flexibility reward, produces significantly shorter yet more effective reasoning, as shown in ablation studies of Table 2.
> - For humans: The human study mirrors this finding. When presented with different reasoning styles, humans rated cognitive flows as significantly more "efficient" than R1's reasoning chains (Figure 4a). This superior structure and efficiency also led to a higher accuracy improvement in human decision-making (Figure 4b).
>
> Therefore, the human study is integral to our argument. It provides evidence that Cognitive Reasoning is a more efficient and effective paradigm for social analysis, which in turn validates its use as the foundation for our CogFlow framework.
>
> ### **Q-3: What degree the LLM evaluator represents human preference in social reasoning situations?**
>
> **A-3:** As the reviewer noted in Weakness 6, *"social reasoning in many situations is by nature ambiguous and may be impacted by personal background and preference"*. Given this high degree of subjectivity, perfect agreement is close to impossible, even among human experts. In fact, the human preference labels used for our comparison were determined by majority vote, which itself implies that disagreement existed even within our expert group. Therefore, achieving a "moderate" kappa score in such a complex, subjective domain is a non-trivial result. It indicates that our LLM evaluator, while not perfect, captures a relatively reliable signal of human preference.
>
> Furthermore, given that large-scale human annotation for RL training and evaluation is computationally and economically impractical, it is essential to build a reliable automated evaluator to serve as a proxy. The purpose of Table 1 was not to find a perfect replacement for human judgment, but to select the best-performing and most human-aligned automated evaluator among several candidates. The results clearly showed that our chosen two-stage method, CPRank$^2$-R1/Q32B, is substantially more aligned with human preferences than all other baselines. This showed that while no automatic evaluator can perfectly mirror human subjectivity, our CPRank$^2$-R1/Q32B method is the most reliable and validated proxy available for this task.
>
> ### **Q-4: In section 4.4, the concept of reasoning depth is used before being properly defined.**
>
> **A-4:** Thanks for pointing out this lack of clarity. In our analysis, the reasoning depth refers to the sequential position of a cognitive unit within the generated cognitive flow. For example, in the flow `Observation -> Attribution -> Behavior`, `Observation` is at depth 1, `Attribution` is at depth 2, and `Behavior` is at depth 3. We will add a clear, formal definition of "reasoning depth" in our revision.

---

> ### Author Response · Authors · 2025-11-21
> **Response to Reviewer 4JhK (3/4)**
>
> ### **Q-5: The human subject study details and ethics.**
>
> **A-5:** Our study received approval from the Institutional Review Board (IRB). As stated in the "Human Participant Engagement" paragraph of the “Ethics Statement” section, our study recruited 6 domain experts (with a master’s degree or higher) for data validation, 6 annotators for evaluating reasoning chains, and 20 volunteers for a cognitive intervention trial. The 6 domain experts were recruited from the psychology department of a top-tier university in the country. The 6 annotators and 20 volunteers were recruited through a crowdsourcing platform. All participants were fairly compensated for their time and contribution based on the market price for such tasks. We balanced the demographics (e.g., gender) of the experts, annotators, and volunteers, and their ages ranged from 18 to 35. All of them were familiar with the concept of LLMs but were not otherwise involved in this research project. To prevent bias, participants were not informed of the specific experimental design, such as the source of the different example instances. They were only informed of the nature of their task (i.e., "evaluate the quality of reasoning" or "make a decision in a social situation").
>
> Regarding our safety filtering process, our mechanism was adapted from Kim et al., 2023 [1]. We applied a rigorous multi-stage process:
> - Canary Filter: First, we employed Canary, a safety classification model, to remove social situations marked as needing caution (e.g., those involving crime).
> - LLM Filter: Next, we prompted DeepSeek-R1 to identify and remove any remaining social situations that exhibited unsafe content or tendencies.
> - Expert Review: Finally, during the data validation process, 6 domain experts were also tasked with reviewing the safety of the social situations, which resulted in a 100% pass rate.
>
> [1]  Hyunwoo Kim, Jack Hessel, Liwei Jiang, Peter West, Ximing Lu, Youngjae Yu, Pei Zhou, Ronan LeBras, Malihe Alikhani, Gunhee Kim, Maarten Sap, and Yejin Choi. Soda: Million-scale dialogue distillation with social commonsense contextualization, 2023. URL https://arxiv.org/abs/2212.10465.
>
> ### **Q-6: The rationale of majority voting.**
>
> **A-6:** This is an insightful point that touches on the core difficulty of this task. Our rationale for using majority voting was to establish a consensus-based preference. In a domain with high ambiguity, relying on a single judge can introduce significant noise from individual preferences. By aggregating the judgments of multiple experts, we aimed to identify the response that was most likely to be preferred, thereby creating a more robust "ground truth" for two specific purposes: 1) validating LLM evaluators aligned with the human consensus (Table 1) and 2) measuring the helpfulness of cognitive flow against the ground-truth response (Figure 4b & 4c).
>
> We agree that using majority voting masks the rich individual differences and the inherent ambiguity of social situations. However, this was a necessary trade-off between establishing a stable benchmark for our experiments and capturing the full spectrum of individual differences. Analyzing the disagreement among annotators, rather than just the consensus, is a promising research direction. It could reveal which types of social situations are most contentious to individual interpretation, and we leave this as an important area for future work.
>
> ### **Q-7: The presentation format as a confounded variable.**
>
> **A-7:** We apologize for not clarifying our experimental controls clearly in the paper, as we took specific steps to mitigate these exact confounding variables in our human evaluations.
>
> **(1) On Presentation Format:** In our human evaluation study (Section 4.2, Figures 4a & 4b), we did not show the structural unit tokens (e.g., `<Observation>`, `<Attribution>`) to the annotators. The cognitive flow was presented as a single, continuous block of plain text, making its format identical to the R1’s reasoning chain. This ensures that annotators were comparing the content of the reasoning, not a structural presentation difference.
>
> **(2) On Reasoning Length:** While our CogFlow method produces more concise reasoning on average, for the 100-instance subset used in this specific human study, we deliberately sampled examples where the cognitive flow and the R1 reasoning chain had comparable lengths. This was done to prevent annotators from simply preferring a shorter or longer chain, thus isolating the effect of the reasoning content itself.
>
> **(3) On the Cognitive Intervention Trial:** This same control for bias was applied to our cognitive intervention trial (Figure 4c). As the examples in Table 8 (Page 28) illustrate, the guidance provided for both the "Cognitive Flow Style" and the "R1 Reasoning Chain Style" was carefully crafted to be comparable in length, structure, and descriptive detail.

---

> ### Author Response · Authors · 2025-11-21
> **Response to Reviewer 4JhK (4/4)**
>
> ### **Q-8: Contradiction in motivation and method.**
>
> **A-8:** We would like to clarify that there is no contradiction between our motivation and our method, and their relationship is as follows.
>
> First, our motivation is that real-world social situations are inherently ambiguous and rarely yield a definitive outcome. This refers to the nature of the problem we are tackling.
>
> Second, our framework (CogFlow) is a pragmatic approach rooted in social cognitive theory to navigate this ambiguity. We do not claim to solve the ambiguity or find a definitive "ground truth".
>
> Third, the reviewer notes our framework "clearly defines those cues and outcomes via heuristics and powerful LLMs". We would like to point out that we are not defining definitive outcomes. Instead, our work defines a structured process that uses LLMs to generate and evaluate a diverse set of potential reasoning paths (i.e., cognitive flows) and their resulting responses. Then, we rely on comparative preference, rather than definitive "ground truth", to guide our data filtering and model training pipeline. This is the mechanism that allows us to operate in an ambiguous space. Specifically:
> - Our "Dual-Validation based Filtering" (Section 3.1) uses "Comparative Preference Ranking" to identify flows that lead to preferred responses, not definitive correct ones.
> - Our RL reward $\mathcal{R}_{Res}$ (Section 3.3) is trained on pairwise preference (which response is better than another), again avoiding the need for a single ground truth.
>
> In summary, our method does not "clearly define" a single correct solution. Instead, it provides a principled framework to find, distill, and teach human-aligned reasoning strategies based on comparative preferences, which is the practical approach in a domain that lacks "definitive answers."

---

> > ### Comment · Reviewer_4JhK · 2025-11-24
> >
> > I thank the authors for the thorough rebuttal. It is well written with rigorous logic and solid supplementary experimental results. I recommend incorporating those clarifications into the paper revision. Since all my concerns are resolved, I have increased my overall rating to 8 and advocate for the acceptance of this paper.

---

> > > ### Author Response · Authors · 2025-11-25
> > > **Response to 4JhK**
> > >
> > > We sincerely thank the reviewer for the time and positive re-evaluation of our work. We are delighted that our rebuttal successfully resolved the reviewer's concerns. We deeply appreciate the reviewer's advocacy for the paper's acceptance. As recommended, we will ensure that all clarifications and supplementary experimental results discussed during the rebuttal are carefully incorporated into the final revision of the paper. Thanks again for the reviewer's constructive feedback, which has significantly strengthened our work.

---

### Official Review · Reviewer_UFD7 · 2025-10-30

**Soundness:** 1
**Presentation:** 2
**Contribution:** 1
**Rating:** 2
**Confidence:** 5

**Summary:**

This paper introduces CogFlow, a pipeline that distills social situations from Reddit using R1, simulates the flow of thoughts through six tagged “cognitive units,” ranks and filters these units with LLM judges, and performs SFT warm-up followed by GRPO on a smaller-scale 8B Qwen model. LLM-as-judge evaluation results show that the CogFlow pipeline yields better reasoning quality and more concise reasoning traces.

**Strengths:**

The paper is well written and easy to read. The targeted domain of social reasoning is interesting and deserves greater attention from the research community.

**Weaknesses:**

- Lack of technical contribution. This paper is one among many recent works that simply adapt SFT + GRPO to a specific domain — in this case, a narrow “social reasoning” setting. Although the authors repeatedly use the term “cognitive,” the work has little to do with actual cognitive science. It is unclear what novel insight or methodological advance this paper offers. Even if the approach generalizes to other tasks (considering broader impact), it seems that the gains would primarily stem from SFT + GRPO, rather than from the proposed “cognitive units.”

- Circluated evaluations. All the questions were generated by a language model (DeepSeek R1). The teacher traces were also generated by a language model, and the student is a language model as well. The performance is evaluated using a language-model-based reward, yet the human agreement (κ) is not outstanding. It’s hard to say whether this setup captures human social cognition or merely reflects the reasoning patterns favored by DeepSeek R1.

- Flawed baselines. Some prompt-engineering baselines are missing — for instance, how would a tuning-free prompt template that enforces a compact, fixed section structure perform? Even simple few-shot prompting using the teacher traces could serve as a meaningful comparison.

- Narrow domain. There are no out-of-domain or cross-task generalization tests. All the distilled traces are also on the same task. It’s hard to say how this would generalize as a universal tool for language model social reasoning.

**Questions:**

Can you think of other tasks your framework can generalize to?

---

> ### Author Response · Authors · 2025-11-21
> **Response to Reviewer UFD7 (1/3)**
>
> We thank the reviewer's time and feedback. However, we respectfully disagree with the assessment of our contribution and evaluation, which we believe stems from the reviewer’s misunderstanding of our paper's primary claims and results. We would like to address the specific concerns below.
>
> ### **Q-1: Lack of technical contribution.**
>
> **A-1:** We respectfully and firmly disagree with the assertion that our work is "simply adapt SFT + GRPO to a specific domain". This view fundamentally underestimates our core contribution.
>
> **First, our conceptual novelty is the cognitive reasoning paradigm, which is grounded in cognitive science theory.** We identify a critical failure mode in existing LLM reasoning: standard deductive, step-by-step logic is not well-suited for ambiguous social situations and leads to "cognitive rumination" (i.e., over-analyzing and producing erroneous judgments, as shown in Figure 1). We operationalize this reasoning process in social situations into a structured "cognitive flow" of six core units (Observation, Attribution, etc.), which are explicitly grounded in established social cognitive theory. This is not a superficial use of the term "cognitive". It is a formal framework for shifting LLM reasoning from logical deduction to interpretive analysis.
>
> **Second, our methodological novelties are cognitive flow simulation and multi-objective reward function.** Our paper does not claim to use GRPO as the novelty. Instead, we propose the CogFlow framework to instill this new reasoning capability into an LLM. This includes:
> - A cognitive flow simulation for data generation (Section 3.1), which moves beyond classical  Chain-of-Thought to model the associative, tree-structured nature of human social thought.
> - A multi-objective reward function for cognitive flow optimization (Section 3.3), which introduces cognitive flexibility reward for teaching LLM to regulate its own thought process. This reward is precisely what prevents both "pattern collapse" (Figure 5c) and "cognitive rumination" (Figure 1), a problem the reviewer overlooks.
>
> **Third, we would like to highlight that our introduced "cognitive units" are essential for performance.** The reviewer's hypothesis that gains "primarily stem from SFT + GRPO, rather than from the proposed 'cognitive units'" is not valid, as shown in our own ablation studies in Table 2. Table 2 clearly showed that our full CogFlow model significantly outperformed Direct-GRPO (which is exactly SFT+GRPO on responses) and Distilled-R1-GRPO$_{\mathcal{R}_{Len}}$ (which applies RL to standard CoT-style reasoning chains). This empirically confirmed that the performance gains stem precisely from our CogFlow framework, not just the base RL algorithm.
>
> ### **Q-2: Circulated evaluations.**
>
> **A-2:** We appreciate the reviewer's concern about "circulated" pipelines. However, the reviewer should have overlooked the extensive human-in-the-loop validation we performed specifically to mitigate this risk. Our results confirm that the model is not merely mimicking its teacher but generating outputs that humans find better than the teacher model.
>
> **First, we employed human experts to conduct pairwise evaluations of models' responses.** As shown in Figure 3 and Table 7 (page 26), our 8B CogFlow model statistically outperformed its teacher, DeepSeek-R1, in direct human-preference comparisons (51.9% win rate). This proves our model is not "merely reflect the reasoning patterns favored by DeepSeek-R1", but is, in fact, generating responses that humans find more preferred.
>
> **Second, we hired human annotators to conduct a pointwise evaluation of the reasoning process itself.** As shown in Figure 4(a), human annotators rated our cognitive flows as more coherent, efficient, and predictable than R1's reasoning chains.
>
> **Third, we conducted a cognitive intervention trial by recruiting human participants.** As shown in Figure 4(b), humans found our cognitive flows more helpful for their own decision-making. Furthermore, as Figure 4(c) shows, training humans' social thinking with cognitive flow-style guidance significantly improved participants' social decision-making accuracy, while training with the R1 reasoning chain-style ones showed a slight downward trend.

---

> ### Author Response · Authors · 2025-11-21
> **Response to Reviewer UFD7 (2/3)**
>
> ### **Q-3: Flawed baselines.**
>
> A-3: We would like to clarify that our evaluation strategy was designed to compare our trained model against the strongest possible prompting baseline, rather than weaker variations.
>
> **First, our experiments have already included the strongest prompting baseline, `Simulated-CogFlow`.** We utilized `Simulated-CogFlow` as the representative prompt-based baseline. This method prompts the teacher model (DeepSeek-R1) to generate reasoning content following our exact, structured cognitive flow constraints via tree-structured planning. This represents an "upper bound" for prompting, far exceeding the capability of simple few-shot templates. As shown in the human evaluation of Figure 3, our 8B CogFlow model achieved a Bradley-Terry relative strength score comparable to this strong teacher-prompted baseline. This shows that our framework successfully internalizes the cognitive capability into a model <1.5% the size of the teacher, achieving an efficiency that tuning-free prompting (Table 2) and CoT prompting (Table 6, page 26) cannot match.
>
> **Second,** to directly address the reviewer's question about using a "tuning-free prompt template" as baseline, we performed a supplementary experiment using a specifically designed "Cognitive Reasoning Prompting" (CRP). Unlike `Simulated-CogFlow`, which uses tree-structured planning, CRP simply embeds the definitions of our six cognitive units and the logic of cognitive flow directly into the system prompt, instructing the base model to generate a reasoned social response. As shown in the Table below, our trained CogFlow model significantly outperformed these CRP-based baselines. These results reinforce the argument in our Related Work: relying solely on prompting leads to "superficial mimicry" of the format, whereas CogFlow ensures the true internalization of social thinking. The prompts and detailed results will be added to the revision.
>
>
> | Models                   | CPRank²-R1(Overall) | CPRank²-R1(Easy) | CPRank²-R1(Med) | CPRank²-R1(Hard) | CPRank²-Q32B(Overall) | CPRank²-Q32B(Easy) | CPRank²-Q32B(Med) | CPRank²-Q32B(Hard) | Reasoning Length(tokens, ↓) |
> | ------------------------ | ------------------- | ---------------- | --------------- | ---------------- | --------------------- | ------------------ | ----------------- | ------------------ | -------------- |
> | Llama-3.1-70B (CRP)  | 0.0933              | 0.1802           | 0.0733          | 0.0264           | 0.1786                | 0.3207             | 0.1509            | 0.0783             | 305.04         |
> | Qwen-2.5-72B (CRP)   | 0.1119              | 0.2340           | 0.0852          | 0.0167           | 0.2509                | 0.3969             | 0.2213            | 0.1506             | 279.67         |
> | DeepSeek-V3 (CRP)    | 0.1641              | 0.3070           | 0.1400          | 0.0451           | 0.3161                | 0.4198             | 0.3038            | 0.2294             | 269.75         |
> | o3 (CRP)             | 0.1822              | 0.3038           | 0.1469          | 0.0959           | 0.3049                | 0.4148             | 0.2834            | 0.2281             | N/A            |
> | Qwen-3-32B (CRP)     | 0.2713              | 0.3258           | 0.2533          | 0.2347           | 0.5418                | 0.5828             | 0.5449            | 0.4935             | 671.04         |
> | Gemini-2.5-pro (CRP) | 0.3798              | 0.5528           | 0.3129          | 0.2730           | 0.4922                | 0.6073             | 0.4677            | 0.4153             | 564.14         |
> | GLM-4.5 (CRP)        | 0.4024              | 0.5052           | 0.3509          | 0.3392           | 0.6521                | 0.7337             | 0.6344            | 0.5844             | 770.18         |
> | DeepSeek-R1 (CRP)    | 0.5271              | 0.5705           | 0.5027          | 0.5032           | 0.6564                | 0.6912             | 0.6448            | 0.6351             | 995.53         |
> | **CogFlow (Ours, Llama version)**       | 0.5756          | **0.6645**           | 0.5350          | 0.5294       | 0.7828            | **0.8271**         | 0.7908        | 0.7232             | 391.68         |
> | **CogFlow (Ours, Qwen version)**       | **0.6652**          | 0.6404           | **0.7531**      | **0.6015**       | **0.7956**            | 0.8248         | **0.7963**        | **0.7641**         | 347.37         |

---

> ### Author Response · Authors · 2025-11-21
> **Response to Reviewer UFD7 (3/3)**
>
> ### **Q-4: Generalization capabilities of our framework.**
>
> **A-4:** To demonstrate that CogFlow functions as a universal reasoning paradigm rather than a domain-specific heuristic, we conducted supplementary experiments on three out-of-domain datasets: MATH-500 [1] (mathematical reasoning), SocialIQA [2] (social commonsense reasoning, Multi-choice Question), and HiToM [3] (theory of mind benchmark, Multi-choice Question).
>
> **(1) Experiment Setup:** In this experiment, all models use Llama-3.1-8B-Instruct as the backbone model. Our baselines include Llama-3.1-8B-Instruct, both Tuning-free and CoT versions. We compared these against two variations of our framework and the strongest R1-distilled baseline from our main paper:
> - Direct Task Transfer: We evaluated the zero-shot performance of our original CogFlow and Distilled-R1-GRPO$_{\mathcal{R}_{Len}}$ (trained on social situations) directly on the new test sets.
> - Task Adaptation: To test the universality of the new paradigm rather than just the data, we utilized the CogFlow-SFT model (trained purely on the original social cognitive flows) as the initialized policy. We then applied our RL training methodology to the training sets of MATH [4] and SocialIQA. Crucially, we did not generate any domain-specific SFT data for these tasks. This rigorous setting tests whether the "social" reasoning structure learned during SFT can generalize to other domains via RL. This approach yielded CogFlow-adapt (trained on CogFlow-SFT) and Distilled-R1-GRPO-adapt (trained on Distilled-R1-SFT).
>
> **(2) Results:** The results are evaluated using accuracy in three benchmarks. As shown in the table below, while Direct Task Transfer yields marginal gains due to domain shift, the Task Adaptation (CogFlow-adapt) demonstrates very strong generalization. It significantly outperforms standard baselines and the Distilled-R1 method trained on the same data. This confirms that the underlying CogFlow framework is not just a "social reasoning" heuristic, but a robust reasoning paradigm helpful beyond social reasoning scenarios.
>
> **(3) Mechanism of Generalization:** It may seem counterintuitive that social cognitive units improve the model’s mathematical ability. However, our inspection of the outputs finds that CogFlow enforces a structured meta-cognitive scaffolding that standard CoT lacks. For example, `Observation` forces the model to explicitly parse and list all problem conditions before solving; `Attribution` helps identify the underlying mathematical principles or theorems required; `Regulation` acts as a self-correction mechanism, checking intermediate calculation steps to prevent the error propagation often seen in standard CoT. This confirms that our framework can generalize as a universal tool for enhancing reasoning structure, far beyond the initial social domain.
>
> | Model                 | MATH-500      | MATH-500              | SocialIQA    | SocialIQA            | HiToM        | HiToM                |
> | --------------------- | ------------ | -------------------- | ------------ | -------------------- | ------------ | -------------------- |
> |                       | Accuracy (↑) | Reasoning Length (tokens, ↓) | Accuracy (↑) | Reasoning Length (tokens, ↓) | Accuracy (↑) | Reasoning Length (tokens, ↓) |
> | Llama-3.1-8B-Instruct | 0.4220       | -                    | 0.7070       | -                    | 0.1942       | -                    |
> | Llama-3.1-8B-Instruct (CoT)    | 0.4400       | 308.18               | 0.6760       | 249.13               | 0.3344       | 722.64               |
> | Distilled-R1-GRPO     | 0.4420       | 1034.54              | 0.6630       | 399.27               | 0.3571       | 1209.72              |
> | CogFlow (Ours)          | 0.4460       | 500.27               | 0.7042       | 227.76               | **0.3625**       | 412.99               |
> | Distilled-R1-adapt    | 0.4900       | 449.88               | 0.7573       | 348.26               | -            | -                    |
> | **CogFlow-adapt (Ours)**     | **0.5340**       | 304.86               | **0.7867**       | 166.28               | -            | -                    |
>
> [1]  Lightman, H., et al. Let’s verify step by step. In International Conference on Learning Representations, 2024.
>
> [2] Maarten Sap, et al. Social IQa: Commonsense Reasoning about Social Interactions. In EMNLP 2019.
>
> [3] Yufan Wu, et al. Hi-ToM: A Benchmark for Evaluating Higher-Order Theory of Mind Reasoning in Large Language Models. In Findings of EMNLP 2023.
>
> [4] Hendrycks, D., et al. Measuring mathematical problem solving with the MATH dataset. In Conference on Neural Information Processing Systems, 2021.

---

> > ### Author Response · Authors · 2025-11-28
> > **Gently Reminder**
> >
> > Dear Reviewer UFD7,
> >
> > Thank you once again for your valuable comments on our submission. As the discussion phase is approaching its close, we would like to kindly confirm whether we have sufficiently addressed all your concerns. Should there be any remaining questions or areas requiring further clarification, please do not hesitate to let us know. If you are satisfied with our responses, we would greatly appreciate your consideration in adjusting the evaluation scores accordingly.

---

### Official Review · Reviewer_mZiH · 2025-10-31

**Soundness:** 4
**Presentation:** 4
**Contribution:** 3
**Rating:** 8
**Confidence:** 3

**Summary:**

CogFlow introduces a framework for aligning large language models with human social reasoning by grounding their thinking process in cognitive theory. Instead of optimizing only for correct answers, CogFlow models the reasoning path as a structured sequence of six cognitive units—Observation, Attribution, Motivation, Regulation, Efficacy, and Behavior—adapted from Bandura’s social cognitive framework.

The system first uses DeepSeek-R1 to generate raw reasoning traces and then filters and ranks them using comparative preference ranking (CPRank²). Each reasoning flow is scored on three theory-based metrics—coherence, interpretability, and predictability—by LLMs acting as judges. The top-ranked reasoning samples become the reference set for training a process-level reward model (RMϕ) that approximates DeepSeek-R1’s and human-like preference judgments.

During reinforcement learning, CogFlow is fine-tuned with a multi-objective reward combining response quality, cognitive-unit diversity, and reasoning-length regularization. The length reward (RLen) enforces concise but complete reasoning, while diversity (RDiv) promotes balanced use of different cognitive units. The final reward encourages structured, clear, and human-like social reasoning processes.

Compared with baselines such as Direct and Distilled-R1 models, CogFlow achieves higher automatic and human evaluation scores. Human judges moderately agree with the LLM-based rankings (κ ≈ 0.47) and consistently prefer CogFlow’s reasoning for coherence and clarity. CogFlow reasoning also improves human decision accuracy in behavioral tasks, showing that process-level training yields reasoning that is more efficient, interpretable, and aligned with human cognition.

**Strengths:**

- This paper introduces a novel framework inspired by cognitive science to enhance LLMs’ reasoning ability in social situations. By decomposing reasoning into distinct cognitive units, the framework enables fine-grained tracking and optimization of process-level rewards.

- The study systematically compares not only different model architectures but also multiple training paradigms—such as Supervised Fine-Tuning (SFT), Generalized Reinforcement Preference Optimization (GRPO), and distilled reasoning models—demonstrating that CogFlow achieves consistently superior performance under full reward configurations.

- The proposed model defines a comprehensive reward structure that integrates both response-level and process-level rewards. Through a trained reward model and auxiliary cognitive metrics, CogFlow provides more structured and interpretable supervision of the reasoning process.

- Human evaluations are incorporated to validate the alignment between the trained reward model and LLM-as-a-judge assessments, offering partial evidence of cross-validation between machine and human preference signals.

- The paper includes detailed ablation studies on key reward components, such as reasoning-length and diversity rewards, confirming their individual a

**Weaknesses:**

- The paper does not include a direct RLHF baseline where the model is trained on **human preference data**. Although human evaluations are used post hoc to validate the **LLM-as-a-judge** results, all reward models are trained exclusively on **LLM-generated preference labels** (from DeepSeek-R1 and Qwen-3-32B). This design raises concerns about **self-referential training–evaluation circularity**, since both the reward signal and the evaluation depend on similar model families. A more robust comparison would be to train a model directly using human preference data via RLHF and evaluate it using the same CogFlow reward models. This would clarify whether CogFlow’s improvement truly comes from the **cognitive-unit decomposition and process-level reward design**, or merely from alignment with its LLM evaluators.

- The evaluation metrics, while diverse, remain largely **subjective**. Although the framework defines cognitive units such as *Observation* and *Attribution*, their assessment still relies on qualitative judgments (e.g., “coherence” or “interpretability”) scored by humans or LLMs without explicit operational definitions. To strengthen the theoretical contribution, the authors could further formalize what constitutes high-quality reasoning within each cognitive unit. For example, the “Observation” unit could be evaluated by checking whether key situational elements are accurately identified and semantically consistent. While such grounding would require substantial additional work, it would make the framework more **reproducible, interpretable, and empirically falsifiable**, turning subjective evaluations into observable reasoning diagnostics.

**Questions:**

I currently do not have any questions.

---

> ### Author Response · Authors · 2025-11-21
> **Response to Reviewer mZiH**
>
> We sincerely thank the reviewer's thorough analysis, positive assessment, and constructive feedback. We are greatly encouraged by the reviewer's positive evaluation of the paper's soundness, presentation, and contribution. The weaknesses commented on provide a clear and valuable roadmap for strengthening this line of research. We would like to offer the following clarifications in response.
>
> ### **Q-1: The lack of a direct human-preference RLHF baseline and self-referential circularity.**
>
> **A-1:** We agree that a direct RLHF baseline trained on human preference data would be an ideal baseline to compare with. However, we should respectfully clarify that the comparison suggested by the reviewer (standard outcome-based RLHF) does not align with our objective of optimizing the reasoning process.
>
> (1) Standard RLHF typically relies on humans to rate final responses. Our CogFlow framework, however, optimizes the step-by-step cognitive flow. Collecting high-quality human labels for intermediate reasoning steps (process supervision) is a significantly more complex and expensive challenge than standard response-level labeling. Therefore, we adopted the roadmap of training our reward model on LLM-generated preference labels as a scalable solution.
>
> (2) We actively mitigated the risk of "self-referential circularity" through two rigorous human evaluation protocols:
> - Evaluator Alignment. We validated our LLM evaluators against human judgments, as the reviewer noted in the Summary, the results show a moderate but positive agreement. This confirms our reward models serve as valid proxies for human preference.
> - Human Study. Most importantly, our human study experiment (Figure 4b&4c) shows that humans using CogFlow guidance significantly improved their social decision-making accuracy compared to those using R1 reasoning. This result provides strong evidence that CogFlow’s improvements stem from the efficacy of the cognitive decomposition itself, rather than merely overfitting to LLM evaluators.
>
> We agree with the reviewer’s valuable suggestion for training on direct human process data, and we will highlight this as a key direction for future work.
>
> ### **Q-2: The subjectivity of evaluation metrics.**
>
> **A-2:** We appreciate this constructive suggestion. We agree that our framework would be significantly strengthened by more "explicit operational definitions". The core challenge, which motivated this work, is the interpretive nature of reasoning in social situations. Unlike logical deduction, social situations involve ambiguous cues and rarely have a definitive reasoning process. This ambiguity makes purely objective, operationalized evaluation difficult. Our current qualitative metrics (e.g., "coherence," "interpretability") used in cognitive flow pruning (Section 3.1) and human evaluations (Figure 4a) were adopted as a pragmatic first step to manage this ambiguity and identify high-quality reasoning processes.
>
> The reviewer’s suggestion to move toward "observable reasoning diagnostics" is a promising direction for extending our research. Formalizing the explicit operational definitions for each cognitive unit, as the reviewer kindly notes, would "require substantial additional work" and constitutes a significant research challenge in itself. Therefore, we view this as a valuable and high-impact avenue for the next stage of our research, building upon the foundational framework we establish in this paper.

---

> > ### Author Response · Authors · 2025-11-28
> > **Gently Reminder**
> >
> > Dear Reviewer mZiH,
> >
> > Thank you once again for your valuable comments on our submission. As the discussion phase is approaching its close, we would like to kindly confirm whether we have sufficiently addressed all your concerns. Should there be any remaining questions or areas requiring further clarification, please do not hesitate to let us know. If you are satisfied with our responses, we would greatly appreciate your consideration in adjusting the evaluation scores accordingly.

---

### Official Review · Reviewer_FyRV · 2025-11-04

**Soundness:** 3
**Presentation:** 2
**Contribution:** 3
**Rating:** 2
**Confidence:** 5

**Summary:**

This paper introduces a novel paradigm called Cognitive Reasoning (CR), which aims to address the inherent limitations of Large Language Models (LLMs) when dealing with complex social situations. The authors argue that traditional logical reasoning paradigms (e.g., Chain-of-Thought, CoT) are suitable for verifiable questions with clear-cut answers, but fail to cope with the ambiguity, interpretive processes, and non-deterministic outcomes common in social contexts. The CR paradigm mimics human social cognition by structuring the interpretive process into a flow of interconnected cognitive units (e.g., observation, attribution). These units combine adaptively to facilitate effective social thinking and responses.

**Strengths:**

1. The topic is interesting and important. Uncertainty exists in social reasoning scenarios a lot.
2. The authors propose their own theory, cognitive flow, which consists of six core cognitive units. And the model can adaptively select a better cognitive reasoning path in a tree structure. Classic but cool idea!
3. The authors also contribute a generated dataset.
4. They conducted comprehensive experiments to show the superiority of their model.
5. The figures to show transition patterns are nice. We get some helpful insights into what's going on inside the model.

**Weaknesses:**

1. The six core cognitive units do not directly come from the reference Bandura 1986. So the validity of the proposed theory needs to be further examined and clarified. You use nouns to name the six core cognitive units -- why not using verbs? And, "Efficacy" seems to be a principle, not a cognitive process. A rational agent plans effectively and efficiently. But in your definition, "efficacy" is not about effectiveness, but rather the belief in executing a social behavior, which confused me a lot. Same problem when you define "motivation". I think the way you define such cognitive terms is very inaccurate.  This substantially diminishes the paper's significance, since it's a problem with the base.
2. When you craft your dataset, you use R1 all the time, with no human verification.
3. How is the next unit selected? LLM again?
4. I believe there are many papers about social reasoning with LLM nowadays. Some use only textual input, some even use videos. I didn't see any discussion or reference to these papers, nor comparisons with their LLM architectures.
5. "Cognitive reasoning" paradigm is not new and not invented by you. Overclaimed. Please study the literature in depth. What is the core difference between your method and these existing methods?
6. Better to show some qualitative results.
7. Why do you choose to use data from Reddit? Will this make the dataset biased?
8. [L180-181] next cognitive units should be k+1.
9. Why do you choose to use R1, not GPT?
10. What are the "5 major categories and 16 subcategories" in detail?
11. [Table 1] kappa values are not high - which shows the problem of the R1-generated dataset.
12. [Figure 6] what does the red color mean?

**Questions:**

See the above.

---

> ### Author Response · Authors · 2025-11-21
> **Response to Reviewer FyRV (1/4)**
>
> We thank the reviewer for his/her time and detailed feedback. We are glad the reviewer found **the topic "interesting and important" and our idea "classic but cool".** While the overall rating was low, we believe there are some fundamental misunderstandings about our methodology and contributions. We address these points below.
>
> ### **Q-1: The validity and definition of the six cognitive units.**
>
> **A-1:** We would like to clarify that the name and definition of our cognitive units are not arbitrary, but based on solid social cognitive theories. More details are provided below:
>
> **(1) Theoretical Foundation:** Our cognitive units are inspired by and operationalize Bandura's Social Cognitive Theory for a computational framework. They are not intended to be a direct, exhaustive list of every component from Bandura (1986) [1]. Instead, as explicitly detailed in Section 2, we synthesized multiple authoritative theoretical models to operationalize this complex process of social cognition into computable units. For example, "Motivation" is derived from Vroom’s Expectancy Theory (Vroom, 1964) [2] and "Efficacy" is derived from Bandura’s work on Self-Efficacy (Bandura, 1997) [3].
>
> **(2) Definitions of "Efficacy" and "Motivation".**
> - **Efficacy:** Our definition of "Efficacy" (i.e., *"The belief to execute a specific social behavior"*) is intentionally and explicitly based on Bandura's foundational concept of "self-efficacy" (Bandura, 1997). It is a cornerstone of Bandura’s social cognitive theory as the belief in one's capabilities to execute a specific course of action, not a measure of outcome.
> - **Motivation:** Similarly, our definition of "Motivation" (i.e., *"The expectations and value assessments... which provide the drive to act"*) is a standard, widely accepted definition of motivation consistent with foundational work in psychology, such as Vroom's (1964) expectancy theory.
>
> **(3) Nouns vs. Verbs:** We chose nouns (e.g., "Observation," "Attribution") to formalize these units because they represent components of a cognitive map or state in the reasoning process. This aligns with standard conventions in both cognitive science and computer science (e.g., "Memory" or "Attention" mechanisms). Using nouns allows us to clearly model the cognitive flow as structured transitions between defined states, rather than merely describing transient actions.
>
> [1] Albert Bandura et al. Social foundations of thought and action. Englewood Cliffs, NJ, 1986(23-28):2, 1986.
>
> [2] Victor H Vroom. Work and motivation. John Willey & Sons, 1964.
>
> [3] Albert Bandura. Self-efficacy: The exercise of control. Macmillan, 1997.
>
> ### **Q-2: When you craft your dataset, you use R1 all the time, with no human verification.**
>
> **A-2:** We respectfully but firmly disagree with the reviewer's assertion that there was "no human verification", as it is factually incorrect.
>
> Our paper clearly describes a rigorous human verification process. As outlined in Section 4.1 "Datasets" (Lines 264-267), we state *"To validate data quality, we employ six domain experts (with Master's degrees or higher) to inspect 500 random instances, resulting in a 96.8% pass rate that confirms the dataset's satisfying quality."*
>
> Additionally, in Section 4.2, we detailed how *"Six experts perform pairwise comparisons on all 500 seed instances in our test set"* specifically to validate the consistency between our LLM evaluators (used in data filtering) and human judgment.
>
> Therefore, our final dataset's quality was confirmed by qualified human experts’ validation, as we clearly reported in the paper.

---

> ### Author Response · Authors · 2025-11-21
> **Response to Reviewer FyRV (2/4)**
>
> ### **Q-3: How is the next unit selected? LLM again?**
>
> **A-3:** Two stages are involved in the next unit selection, i.e., data generation and model inference. And we would like to clarify the difference between these two stages below.
>
> **(1) During the data generation stage (Section 3.1, "Planning"):** The selection in this stage is performed by the LLM (DeepSeek-R1) in a structured manner. As described in Lines 179-182, we prompt R1 to perform a specific "Prediction" step. In this step, the model analyzes the current reasoning path and predicts a set of relevant, contextually appropriate next cognitive units from the six available options. This mechanism adaptively prunes the action space, ensuring that only logical continuations are explored. It also models the "associative and progressive process of human cognition", allowing the simulation to branch and explore multiple valid reasoning paths rather than following a single, fixed chain.
>
> **(2) During the model inference stage (After Training):** After SFT and RL, our final CogFlow model learns to generate the entire cognitive flow end-to-end. For example, it generates the next cognitive unit (e.g., `<Attribution>`) as a special token, which then guides the generation of the content for that unit. This is shown in our information flow analysis (Figure 6), where the unit tokens (e.g., `</Observation>`) have a strong influence on the generation of the next unit's content.
>
> ### **Q-4: The lack of discussion and comparison with other social reasoning papers.**
>
> **A-4:** We have to respectfully remind the reviewer that our Related Work (Section 5) is dedicated to positioning our work within this growing field of social reasoning. We explicitly discuss and cite numerous recent papers on LLM social behavior (e.g., Park et al., 2023), social cognition (e.g., Dong et al., 2025), theory of mind (e.g., Chen et al., 2024; Wilf et al., 2024; Sarangi et al., 2025), and cognitive-inspired reasoning (e.g., Gandhi et al., 2023; Wang et al., 2024a; Park et al., 2025; Xu et al., 2025).
>
> The core of our argument and comparison is that most existing methods fall into prompt-based reasoning. Our key distinction is that we move beyond prompting to internalize an adaptive, structured reasoning process into the LLM via SFT and RL. This forms a unique type of long chain-of-thought for social reasoning with LLMs. While multimodal (e.g., video) inputs are an interesting future direction, our work focuses on first establishing the computational paradigm for cognitive reasoning in the text domain.
>
> ### **Q-5: What is the core difference between your method and these existing methods?**
>
> **A-5:** We would like to clarify that we did not claim to have invented the psychological concept of cognitive reasoning. Instead, our contribution is introducing **Cognitive Reasoning as a new computational paradigm for LLMs**, which is designed to equip LLMs with a new form of reasoning paradigm  to  deal with social situations. It is on par with the dominant "logical reasoning" paradigm to deal with math, coding, and logical deduction problems. More specifically, we distinguish from existing methods in two key ways.
>
> **(1) Paradigm definition.** Current LLM reasoning paradigms (e.g., chain of thought) rely on logical deduction, which excels at tasks with verifiable answers (e.g., math, coding). We argue this is not best suited for social situations, which require an interpretive process to handle ambiguous cues and non-deterministic outcomes (Lines 40-48). We formalize this process into a structured "cognitive flow" composed of six core cognitive units (Observation, Attribution, etc.), defining a reasoning structure explicitly tailored for social ambiguity.
>
> **(2) Methodological implementation.** As detailed in section 5 "Related Work", our method differs from existing literature in how the capability is induced:
> - Existing methods (prompting mimicry): Most prior work in social reasoning relies on prompting strategies. This often results in "superficial mimicry" of a format, without changing the underlying reasoning mechanics of the model.
> - Our method (capability internalization): Our CogFlow is a complete training framework. By utilizing Supervised Fine-Tuning and Multi-Objective Reinforcement Learning, we internalize this social cognitive capability into the LLM's parameters. This allows the model to adaptively regulate its cognitive flow based on the situation (as evidenced by the structural analysis in Figure 5), rather than rigidly following a static prompt.

---

> ### Author Response · Authors · 2025-11-21
> **Response to Reviewer FyRV (3/4)**
>
> ### **Q-6: Better to show some qualitative results.**
>
> **A-6:** We appreciate the reviewer's emphasis on qualitative analysis. We would like to highlight that the paper currently indeed includes quite some qualitative results designed to illustrate both the output quality and the internal reasoning mechanisms of our model:
> - **Comparative case study (Figure 1):** We present a detailed side-by-side comparison of a complex social situation. This figure directly contrasts the lengthy, ruminative output of "Conventional Reasoning" from the DeepSeek-R1 model with the concise, effective, and structured "Cognitive Reasoning" generated by our model.
> - **Reasoning mechanism visualization (Figures 5 & 6):** We go beyond text outputs to visualize the model's internal dynamics. Figure 5 analyzes the structural transition patterns, showing how the model hierarchically organizes cognitive units. Figure 6 visualizes the information flow via attention weights, confirming that cognitive unit tokens actively steer the content generation process.
> - **Intervention examples (Appendix D.7):** Table 8 provides concrete, full-text examples of the "Cognitive Flow Style" versus "R1 Reasoning Chain Style" guidance. These qualitative samples used in our human intervention study illustrate exactly how the two reasoning paradigms differ in a human-readable format.
>
> ### **Q-7: Why do you choose to use data from Reddit? Will this make the dataset biased?**
>
> **A-7:** We selected Reddit specifically to address the lack of complexity of social situations in current benchmarks. As discussed in Section 3.1, existing datasets like SocialIQA are often limited to simple situations with limited social dynamics (for example, a case from SocialIQA is *"Aubrey never told Riley the answer and Riley was angry"*). To rigorously evaluate social cognitive reasoning, we required data reflecting realistic and complex social situations, for which Reddit is a rich resource.
>
> We were highly conscious of potential bias and mitigated this in three key ways:
> - Diverse sources: We explicitly avoided sourcing from a single Reddit community. As detailed in Appendix D.1, our dataset is drawn from 18 different subreddits (e.g., r/family, r/socialskills, r/FriendshipAdvice, r/relationship_advice), ensuring a wide variety of social contexts.
> - Balanced distribution: As shown in Table 5 (Appendix D.1), the 5,100 situations are distributed across 5 major categories (Professional, Family, Friendship, Public, and Romance) and 16 subcategories.
> - Rigorous filtering: All collected posts underwent a strict processing pipeline. As detailed in the Ethics Statement and Section 4.1, this included complete anonymization to remove all sensitive or personal content and the application of rigorous safety filters to screen all data.
>
> ### **Q-8: [L180-181] next cognitive units should be k+1.**
>
> **A-8:** We sincerely thank the reviewer for catching this typo. We will correct this error in the final manuscript.
>
> ### **Q-9: Why do you choose to use R1, not GPT?**
>
> **A-9:** This work took approximately six months of effort. When we initiated this work, R1 was the state-of-the-art and most prominent open-source reasoning model at that time, which is why we selected it as our primary model for our methodology.
>
> We also would like to emphasize that our methodology is fundamentally model-agnostic and not restricted to any specific model. Additionally, we did conduct extensive comparisons against a wide array of recent popular LLMs. As shown in Table 2 and Table 6 (Appendix D.5) of our paper, our baselines include LLMs from GPT, Qwen, Gemini, GLM, and DeepSeek. These results consistently show that our trained CogFlow model achieved very strong performance, validating the effectiveness of our framework.
>
> ### **Q-10: What are the "5 major categories and 16 subcategories" in detail?**
>
> **A-10:** This category information was provided in full detail in the appendix. As noted at the end of Section 4.1 (Page 5), the complete distribution and detailed list of these categories are available in Appendix D.1, Table 5 (Page 23).
>
> Table 5 in the appendix lists all 5 major categories (Romance, Family, Public, Friendship, Professional) and their 16 respective subcategories (e.g., Dating & Courtship, Parent-Child Interaction, Stranger Encounters, Task-Oriented Collaboration, etc.), along with their precise distribution counts and percentages in our dataset.

---

> ### Author Response · Authors · 2025-11-21
> **Response to Reviewer FyRV (4/4)**
>
> ### **Q-11: [Table 1] kappa values are not high - which shows the problem of the R1-generated dataset.**
>
> **A-11:** We have to respectfully point out that the reviewer's interpretation of Table 1 is incorrect:
>
> (1) Table 1 does not measure the quality of our R1-generated dataset. The table's purpose is to validate the consistency of *LLM evaluators* against human judgment.
>
> (2) Unlike logical reasoning tasks with verifiable answers, social situations involve analyzing ambiguous cues where definitive ground truth rarely exists. In this subjective domain, even human experts often have disagreements. Therefore, achieving "moderate agreement" (kappa values in the 0.41-0.6 range) is an acceptable and positive result, which validates that our proposed CPRank$^2$-R1/32B evaluators are reliable proxies for human judgment.
>
> (3) The quality of our dataset was validated in a separate procedure described in Section 4.1. Six domain experts inspected the data and reported a 96.8% pass rate, confirming that the R1-generated data is of high quality.
>
> ### **Q-12: [Figure 6] what does the red color mean?**
>
> **A-12:** The "red" color is a darker shade of yellow. The "red" lines indicate the attention flow originating from the "unit content" (depicted as yellow blocks). We used this color to visually distinguish how the textual content of a specific unit (e.g., `Content_obser.`) influences the downstream reasoning process, specifically directed towards subsequent cognitive unit tokens (e.g., `<Regulation>`) and future content.

---

> > ### Author Response · Authors · 2025-11-28
> > **Gently Reminder**
> >
> > Dear Reviewer FyRV,
> >
> > Thank you once again for your valuable comments on our submission. As the discussion phase is approaching its close, we would like to kindly confirm whether we have sufficiently addressed all your concerns. Should there be any remaining questions or areas requiring further clarification, please do not hesitate to let us know. If you are satisfied with our responses, we would greatly appreciate your consideration in adjusting the evaluation scores accordingly.

---

### Meta-Review · Area_Chair_Nwom · 2026-01-01

**Summary:**

The authors propose cognitive reasoning for social settings which are different from general LLM usages.

They train a pipeline called CogFlow with cognitive flows for snall LLMs to produce shorter and more strucutred social reasoning.

Reviewers agree that the topics are important, the pipeline is clear, and the experiments are reasonable. Main cocnerns arise on the novelty -- whether this is just another SFT+GRPO on a new data. Also some reviewers are not buying the deep connection to cognitive sicnece. Finally, the evaluation circularity might be an issue.

On the ethical side, they have dataset sourced from Reddit + human subject studies. The authors claim anonymization + multi-stage safety filtering + IRB approval (in rebuttal). Can be better but not a dealbreaker.

**Reviewer Concerns:**

I read it through and find the followings are the most critical:

1. The paper topic can be too broad while the title/scope may face the issue of overclaiming. I think the objective is ambitious while I personally appreciate more toned-down title/claims

2. Some reviewers challenges on the novelty while I am personally feel it is ok. The pipeline is nothing new while putting all these together is indeed novel enough.

3. Major concerns arise on the data construction, potential bias, potential ethical issues, including useing LLM for evaluation in the later stage. Almost all the reviewers want much more indepth discussion about the data and evaluation

**Reviewer Scores:**

FyRV (2, very confident): likely moves up some, but maybe not all the way. Still, the complaints around theoty validity and overclaim are there. 2->4 is my best guess.

UFD7 (2, very confident): similarly -- I believe the ablation helps while more likely move from 2->4 since the reviewer has strong mind on the technical contribution.

The other two will either stay as 8 or move to 8, which are quite positive.

---

### Decision · Program_Chairs · 2026-01-26

Reject